# *Collapse-Proof* Non-Contrastive Self-Supervised Learning

**Emanuele Sansone** [1 2]   **Tim Lebailly** [1]   **Tinne Tuytelaars** [1]

## Abstract

We present a principled and simplified design of the projector and loss function for non-contrastive self-supervised learning based on hyperdimensional computing. We theoretically demonstrate that this design introduces an inductive bias that encourages representations to be simultaneously decorrelated and clustered, without explicitly enforcing these properties. This bias provably enhances generalization and suffices to avoid known training failure modes, such as representation, dimensional, cluster, and intracluster collapses. We validate our theoretical findings on image datasets, including SVHN, CIFAR-10, CIFAR-100, and ImageNet-100. Our approach effectively combines the strengths of feature decorrelation and cluster-based self-supervised learning methods, overcoming training failure modes while achieving strong generalization in clustering and linear classification tasks.

## 1. Introduction

Self-supervised learning (SSL) has unlocked the potential of learning general-purpose representations from large amounts of unlabeled data. Despite its successes, important challenges remain, hindering the applicability of SSL to a broader spectrum of real-world tasks and its widespread adoption and democratization. One such challenge is the presence of failure modes occurring during the training of SSL models. Several heuristic strategies have been proposed and analyzed in the literature, such as momentum encoder, stop gradient and asymmetric projector heads (Chen et al., 2022; He et al., 2020; Grill et al., 2020; Tao et al., 2022; Chen & He, 2021; Tian et al., 2021; Halvagal et al., 2023; Wang et al., 2022). However, these heuristics do not always come with universal guarantees, making it unclear whether failure modes can be avoided in all situations.

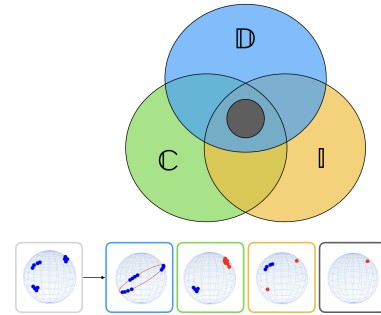

Figure 1: **Top:** Venn diagram relationships between types of collapses, viz. dimensional collapse ($\mathbb{D}$), cluster collapse ($\mathbb{C}$), intracluster collapse ($\mathbb{I}$) and representation (full) collapse of embeddings. **Bottom:** on the left, *collapse-proof* data embeddings (each blue dot correspond to an embedding of a data point, e.g. an image) and on the right, examples of collapses for each single type (collapsed clusters are highlighted in red).

In this work, we focus on the family of non-contrastive SSL approaches, aiming to distill the essential principles to guarantee the avoidance of known failure modes. More concretely, we identify sufficient conditions to avoid representation, dimensional, cluster, and intracluster collapses, exemplified in Figure 1, and correspondingly devise a projector and loss function enforcing them by design. In particular, we demonstrate that minimizing invariance to data augmentations while matching priors suffices to avoid representation and cluster collapses, whereas orthogonal frozen weights based on hyperdimensional computing, and large prediction outputs in the projector are key to avoiding dimensional and intracluster collapses. Moreover, we prove that these principles are sufficient to guarantee (i) the decorrelation of embeddings, without any explicit computation of their covariance matrix, and (ii) the clustering of embeddings, without the use of specialized clustering layers. We experimentally validate the theory on four image datasets, including SVHN, CIFAR-10, CIFAR-100 and ImageNet-100, showcasing training robustness to failure modes and strong generalization to downstream clustering and classification tasks compared to popular feature decorrelation and cluster-based SSL.

To summarize, our key contributions are:

[1]Department of Electrical Engineering (ESAT), KU Leuven, Belgium [2]CSAIL, MIT, US. Correspondence to: Emanuele Sansone <emanuele.sansone@kuleuven.be>.

*Proceedings of the $42^{nd}$ International Conference on Machine Learning*, Vancouver, Canada. PMLR 267, 2025. Copyright 2025 by the author(s).

- At a conceptual level, we identify sufficient conditions to avoid known failure modes and accordingly devise *CPLearn*, the first **C**ollapse-**P**roof non-constrastive self-supervised **Learn**ing approach.

- These conditions are shown to be sufficient to jointly decorrelate and cluster embeddings, thus providing evidence on the feasibility of unifying non-contrastive and cluster-based SSL families of approaches.

- We establish the first connection between SSL and hyperdimensional computing, thus supporting training and the systematic exploitation of large projector outputs.

- At a practical and computational level, we simplify the design and training of non-contrastive SSL and provide a proof-of-concept demonstration of the properties of *CPLearn*.

The structure of the paper is organized as follows: In §2, we relate our work to theory of SSL, clarify the relation between different failure modes and review recent efforts aiming to address them. In §3, we provide design principles for the loss and projector head of *CPLearn*. Subsequently, we provide the theory supporting its design. In §4, we compare our solution to existing feature decorrelation and cluster-based SSL strategies and analyze the different properties highlighted by the theory. Finally, we conclude with §5 by summarizing the main findings and discussing future work.

## 2. Related Work

We frame this work within the context of theoretical studies of SSL and solutions aimed at mitigating collapses.

**Theory and relations among different families of SSL**. Several works have theoretically investigated contrastive (Saunshi et al., 2019; Wang & Isola, 2020; Zimmermann et al., 2021; Tosh et al., 2021; HaoChen et al., 2021; Saunshi et al., 2022; Wang et al., 2024) and non-contrastive SSL methods (Tian et al., 2021; Kang-Jun et al., 2022; Weng et al., 2022; Wen & Li, 2022; Shwartz-Ziv et al., 2023), to improve our understanding and provide more principled or simplified solutions. There have been works identifying key properties of SSL objectives and inductive biases (Wang & Isola, 2020; Dubois et al., 2022), generalizing SSL to incorporate data augmentation graphs (HaoChen et al., 2021; Wang et al., 2024), deriving generalization error bounds (Saunshi et al., 2022; Bao et al., 2022; Shwartz-Ziv et al., 2023), understanding SSL objectives from an optimization perspective (Tian et al., 2021; Tian, 2022; 2023) as well as understanding the role of asymmetries and projector heads (Kang-Jun et al., 2022; Wen & Li, 2022). A recent line of studies has focused on identifying connections between contrastive and non-contrastive methods (Garrido

et al., 2023b; Balestriero & LeCun, 2022; Huang et al., 2023) aiming towards unifying different families of SSL. Our work complements these efforts by providing a principled solution and design to bring together cluster-based SSL and feature decorrelation methods from the non-contrastive family.

**Failure modes in SSL**. SSL can be affected by four undesired failure modes, namely representation, dimensional, cluster and intracluster collapses, as exemplified in Figure 1. *Representation collapse* refers to the case where neural representations collapse to an identical constant vector, irrespectively of their input. Different strategies have been proposed to avoid the issue, such as leveraging contrastive objectives to maximize mutual information between data and representations (den Oord et al., 2018; O. Henaff, 2020; Chen et al., 2020; Lee, 2022; Linsker, 1988; Becker & Hinton, 1992; McAllester & Stratos, 2020; Barber & Agakov, 2004; Belghazi et al., 2018; Poole et al., 2019; Tschannen et al., 2019; Song & Ermon, 2020), introducing heuristics such as momentum encoder, stop gradient and asymmetric projector heads (Chen et al., 2022; He et al., 2020; Grill et al., 2020; Tao et al., 2022; Chen & He, 2021; Tian et al., 2021; Halvagal et al., 2023; Wang et al., 2022), regularizing the objective by introducing a generative term to reconstruct or estimate the data density (Hendrycks et al., 2019; Winkens et al., 2020; Mohseni et al., 2020; Kim & Ye, 2022; Gatopoulos & Tomczak, 2020; Zhue et al., 2020; Sansone & Manhaeve, 2022; Wu et al., 2023; Nakamura et al., 2023; Sansone, 2023; Sansone & Manhaeve, 2023; 2024) and leveraging predictive models to identify masked data such as image patches or text tokens (Assran et al., 2022; Zhou et al., 2022). *Dimensional collapse* occurs when embeddings span a subspace of the whole vector space. Several methods (Zbontar et al., 2021; Zhang et al., 2021; Ermolov et al., 2021; Li et al., 2022b; Liu et al., 2022; Bardes et al., 2022a;b; Ozsoy et al., 2022) propose to mitigate the issue by whitening the feature embeddings (Hua et al., 2021). Dimensional collapse has been recently linked to reduced performance of downstream tasks (He & Ozay, 2022; Li et al., 2022a; Garrido et al., 2023a), and different evaluation metrics have been consequently derived, such as the computation of the entropy of the singular value distribution for the covariance matrix (Jing et al., 2022), the rank estimator (Garrido et al., 2023a), the computation of the AUC (Li et al., 2022a) or a power law approximation (Ghosh et al., 2022) of the singular value distribution. *Cluster collapse* is a phenomenon observed in cluster-based SSL, where data points are assigned to a subset of available prototypes (Caron et al., 2018; Asano et al., 2020; Caron et al., 2020; Li et al., 2021; Caron et al., 2021; Govindarajan et al., 2023). The issue is typically mitigated by introducing regularizers in the objective, such as the Koleo regularizer (Sablayrolles et al., 2019; Oquab et al., 2024; Govindarajan et al., 2024) or explicitly enforcing uniform cluster assignments (Amrani et al.,

2022; Assran et al., 2023). Last but not least in terms of importance, *intracluster collapse*, similarly to the notion of neural collapse observed in supervised learning (Papyan et al., 2020; Fang et al., 2021; Yang et al., 2022; Chen et al., 2022; Kothapalli, 2023; Dhuliawala et al., 2023) occurs whenever the variability of the embeddings within some clusters is infinitesimally small. Intracluster collapse can be mitigated by enforcing representation equivariance (rather than invariance) to data augmentations (Dangovski et al., 2022; Komodakis & Gidaris, 2018; Scherr et al., 2022; Park et al., 2022) or by splitting the embeddings into content and style parts, while using only content for the self-supervision task (Louizos et al., 2016; Kügelgen et al., 2021; Garrido et al., 2023c). In contrast, this work provides a principled yet simple solution based on an objective function and projector head that avoid all forms of collapses.

## 3. Method and Properties

**Notation.** We denote matrices using capital bold letters, e.g. $\mathbf{P}$, their elements using lowercase letters with subscript indices, e.g. $p_{ij}$, their row and column vectors using lowercase bold letters, e.g. $\mathbf{p}_i$ and $\mathbf{p}_j$. Additionally, we denote sets using capital letters, e.g. $\mathcal{S}$ and use squared brackets when dealing with sets of integers, e.g. $[n] \equiv \{1, \ldots, n\}$. Finally, we use lowercase letters for functions, scalars, integers and constants, e.g. $n$. Whenever evaluating functions on matrices, we always assume that the function is applied row-wise.

**Overview of *CPLearn*.** Given an unlabeled batch of data $\mathcal{D} = \{(\mathbf{X}, \mathbf{X}')\}$ containing $n$ pairs of augmented images, so that $\mathbf{X}, \mathbf{X}' \in \mathbb{R}^{n \times d}$, we propose to train a backbone encoder $g : \mathbb{R}^d \to \mathbb{R}^f$ using the *CPLearn* projector and loss functions.[1] The projector takes the representations $(\mathbf{Z}, \mathbf{Z}') = (g(\mathbf{X}), g(\mathbf{X}'))$, with $\mathbf{Z}, \mathbf{Z}' \in \mathbb{R}^{n \times f}$, and performs two operations. Firstly, it computes embeddings $\mathbf{H}, \mathbf{H}' \in \mathbb{R}^{n \times f}$ for the corresponding representations and then it computes probabilities $\mathbf{P}, \mathbf{P}' \in \mathbb{R}^{n \times c}$ for assigning embeddings to codes available from a frozen dictionary $\mathbf{W} \in \{-1, 1\}^{f \times c}$. More precisely, the projector is defined by the following two layers:

$$\mathbf{H} = \sqrt{f/n} \cdot \text{L2-norm}(\text{Bn}(\text{Linear}(\mathbf{Z})))$$
$$\mathbf{P} = \text{Softmax}(\mathbf{HW}/\tau) \tag{1}$$

where the embeddings are obtained from representations through the composition of linear, batch norm and L2 normalization layers, and $\tau$ is the temperature parameter of the softmax layer. Each element $w_{ij}$ of $\mathbf{W}$ is drawn independently and identically distributed according to a Rademacher distribution, i.e. $w_{ij} \sim$ Rademacher for all $i \in [f]$ and

[1] For images $d$ is the product of the width, height and color bands.

$j \in [c]$, whereas

$$\tau = \frac{f}{\sqrt{n} \log\left(\frac{1 - \epsilon(c-1)}{\epsilon}\right)}$$

with $\epsilon$ an arbitrarily small positive scalar.[2] Notably, we will provide theoretical justification for the design choice of the *CPLearn* projector, demonstrating good properties for the embeddings, as being both well clustered and having their features decorrelated. Moreover, we suggest to choose $c \gg f$ to avoid dimensional and intracluster collapses, as we will show in §3.2 and §3.3. The *CPLearn* loss consists of two terms, including one to promote invariance to data augmentations and one for prior matching, namely:

$$\mathcal{L}_{CPLearn}(\mathcal{D}) = -\frac{\beta}{n} \sum_{i=1}^{n} \sum_{j=1}^{c} p_{ij} \log p'_{ij}$$
$$- \sum_{j=1}^{c} q_j \log \frac{1}{n} \sum_{i=1}^{n} p_{ij} \tag{2}$$

with $\mathbf{q} = [q_1, \ldots, q_c] \in \mathbb{Q}^c$ corresponding to a prior probability vector, chosen uniformly for all $c$ codes in all our experiments, viz. $q_j = 1/c$ for all $j \in [c]$, $\beta > 0$ is a weight hyperparameter to balance the relative importance of the two loss terms and $p_{ij}, p'_{ij}$ are elements of $\mathbf{P}, \mathbf{P}'$, respectively. We will prove in §3.1 that, when the two loss terms are minimized, the proposed loss function is guaranteed to avoid representation and cluster collapses and therefore allows to train both the backbone and the projector networks through backpropagation without requiring any additional heuristics, such as stop gradient, momentum encoder or clustering operations typically introduced in non-contrastive learning (Chen et al., 2022; He et al., 2020; Grill et al., 2020; Tao et al., 2022; Chen & He, 2021; Tian et al., 2021; Halvagal et al., 2023; Wang et al., 2022). Taken altogether, the *CPLearn* projector and loss functions guarantee to train the backbone network in a robust manner, preventing all known forms of collapses. Consequently, this represents the first *collapse-proof* non-contrastive SSL solution and we summarize the method together with its PyTorch-like pseudo-code in Figure 2.

### 3.1. Minima of the Loss Function

In the following, we assume that the backbone has infinite capacity. Further discussion about the relaxation of this assumption is left to Appendix C. Therefore, we decouple the study of the objective and its minima from the neural network. In Appendix D, we prove that

[2] $\epsilon = 1e - 8$ throughout the paper.

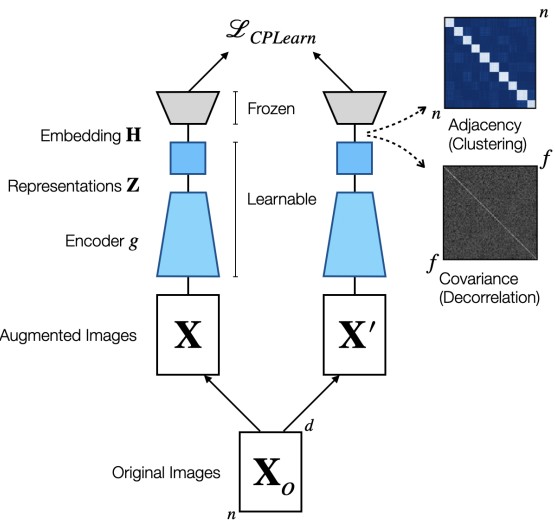

Figure 2: In *CPLearn*, minimizing the proposed objective together with the corresponding projector ensures that the embedding representations are clustered and at the same time that their features are decorrelated. This guarantees that the representations are *collapse-proof*, meaning that dimensional, cluster, intra-cluster and representation collapses are prevented.

**Algorithm 1** Pseudocode for *CPLearn*

```
# g: encoder network
# n: batch size
# f: embedding dimensionality
# c: dictionary size
# eps: 1e-6
# bn: batch normalization
# norm: L2 normalization activation
# softmax: softmax activation with temperature
# beta: weight for the invariance  loss
# ce: crossentropy loss

# compute non-learnable dictionary codes
W = 2 * randint(2, size=(f, c)) - 1 # f-by-c
for X_o in loader: # load a batch with n samples
    # two randomly augmented versions of X_o
    X, X' = augment(x_o)
    # compute representations
    Z = g(X)   # n-by-f
    Z'= g(X') # n-by-f

    # extract embeddings (blue square block)
    H = norm(bn(linear(Z))) * sqrt(f / n)  # n-by-f (*)
    H'= norm(bn(linear(Z'))) * sqrt(f / n) # n-by-f (*)
    # compute probabilities (gray block)
    tau = f / (sqrt(n) * log((1 - eps * (c - 1)) / eps))
    P = softmax(H @ W, tau)  # n-by-c
    P'= softmax(H' @ W, tau) # n-by-c
    # compute losses
    loss_prior = ce(1 / c, P.mean(0))
    loss_inv = ce(P, P').mean()
    loss = beta * loss_inv + loss_prior

    # optimization step
    loss.backward()
    optimizer.step()
```

(*) See Practical Considerations for a simpler alternative to L2-normalization

---

**Lemma 1** (Minima). $\forall i \in [n]$ and $j \in [c]$, $\epsilon \leq p_{ij} \leq 1-\epsilon(c-1)$ and $\epsilon \leq q_j \leq 1-\epsilon(c-1)$ with $0 \leq \epsilon < 1/c$, then the global minima of the loss function in Eq. 2 jointly satisfy the following conditions:

- **Invariance** $\forall i \in [n]$ and $j \in [c]$, $p_{ij} = p'_{ij}$.

- **Extrema** $\forall i \in [n]$, $\exists! j \in [c]$ such that $p_{ij} = 1 - \epsilon(c-1)$ and $\forall k \in [c]$ with $k \neq j$, $p_{ik} = \epsilon$. Here, the word extrema refers to the extrema of a probability simplex.

- **Matched prior** $\forall j \in [c]$, $\frac{1}{n}\sum_{i=1}^{n} p_{ij} = q_j$. Moreover, $\forall j \in [c]$ define $I_{max}(j) \equiv \{i \in [n] : p_{ij} = 1 - \epsilon(c-1)\}$, then $|I_{max}(j)| = \left(\frac{q_j - \epsilon}{1 - c\epsilon}\right) n$.

The global minimum value of Eq. 2 is bounded by

$$\mathcal{L}_{CPLearn}(\mathcal{D}) \geq -\beta(1 - \epsilon(c-1))\log(1 - \epsilon(c-1))$$
$$- \beta\epsilon(c-1)\log\epsilon + H(\boldsymbol{q}) \qquad (3)$$

being equal to $H(\boldsymbol{q})$ whenever $\epsilon = 0$, where $H(\boldsymbol{q})$ is the entropy of $\boldsymbol{q}$.

---

The assumptions in the Lemma can always be met by properly choosing arbitrarily small $\epsilon$ to satisfy the relation $0 \leq \epsilon < 1/c$. The results of the Lemma can be intuitively explained by observing that the first two conditions (invariance and extrema) and the last one (matched prior) are mainly a by-product of the invariance and the matching prior losses in Eq. 2, respectively. Indeed, note that the invariance loss can be equivalently expressed as a cross-entropy loss. Therefore, it can be decomposed into the sum of an entropy term for $\boldsymbol{p}_i$ and a Kullback-Leibler (KL) divergence term between $\boldsymbol{p}_i$ and $\boldsymbol{p}'_i$, thus enforcing the extrema condition through the entropy term and the invariance condition through the minimization of the KL one. Minimizing the matching prior loss is equivalent to minimize the KL between $q_j$ and $1/n \sum_{i=1}^{n} p_{ij}$, thus enforcing the matched prior condition. It is also important to specify that while the results of the Lemma are general and valid for any prior distribution $\boldsymbol{q}$, our focus is mainly on the uniform setting. We leave the study of the non-uniform case (Assran et al., 2023) to future work.

An important implication of the Lemma is that the global minima of the *CPLearn* objective guarantee to avoid representation and cluster collapses, consequently reducing the need for heuristics, such as stop gradients, momentum encoder and/or specialized clustering layers typically introduced in non-contrastive settings. Indeed, we observe that all data points indexed by $i \in [n]$ are assigned in a hard way to one of the codes available in the dictionary due to the extrema condition. Moreover, the distribution of the assignments follows the result of the matched prior

condition, specifically $|I_{max}(j)| = n(q_j - \epsilon)/(1 - c\epsilon)$. For a uniform prior $q_j = 1/c$, we have that $|I_{max}(j)| = n/c$ for all codes $j$, meaning that data points are partitioned and equally assigned to all available codes. Representation collapse is prevented because data points are assigned in a hard fashion to different codes, whereas cluster collapse is avoided because all codes contribute to the partitioning of the data. Moreover, attaining the lower bound value in Eq. 3 gives a certificate for the avoidance of these collapses.

A similar guarantee result has recently appeared in another work (Sansone, 2023), where the same training objective to Eq. 2 is used in addition to a likelihood-based generative term. Their analysis studies each loss term separately, demonstrating their different properties. That is, the generative term prevents representation collapse, the invariance term enforces smoothness and adherence to the cluster assumption, while the matching prior loss prevents cluster collapse. Differently from (Sansone, 2023), we study the objective where the invariance and the matching prior losses are jointly optimized and show through Lemma 1 that they are sufficient to avoid the two collapses without the need of additional terms, like the generative one used in (Sansone, 2023).

### 3.2. Properties of the Projector

We now turn the analysis to the design of the projector and state the main theorem of this work for the case of $c = f$ (the proof is provided in Appendix E). We will later see how to generalize these results to $c \neq f$. Importantly, the theorem sets the stage for the two key properties of *CPLearn*, that is of learning decorrelated and clustered features and avoiding dimensional and intracluster collapses.

---

**Theorem 1** (Embedding). *Given the projector defined in Eq. 1 with $c = f > 2$ and a dictionary matrix $\boldsymbol{W}$ satisfying the condition $\boldsymbol{W}^T \boldsymbol{W} = f\boldsymbol{I}$, if the optimality conditions of Lemma 1 are met, then the embeddings $\boldsymbol{H}$ satisfy the following relation*

$$\forall i \in [n], \exists! j \in [c] \; s.t.$$
$$\boldsymbol{h}_i = \alpha_{ij}\boldsymbol{w}_j + \left(\alpha_{ij} - \frac{1}{\sqrt{n}}\right)\sum_{k \neq j}\boldsymbol{w}_k \quad (4)$$

*with $\alpha_{ij} \in \left\{\frac{1}{\sqrt{n}}, \left(1 - \frac{2}{c}\right)\frac{1}{\sqrt{n}}\right\}$.*

---

The theorem tells that at optimality embeddings align with the orthogonal codes from the dictionary. More concretely, each embedding aligns with one code up to some spurious additive term, i.e. the second addend in Eq. 4, whose contribution depends on the admissible values of the coefficient $\alpha_{ij}$ and the remaining codes. Notably, if $\alpha_{ij} = 1/\sqrt{n}$, the spurious term disappears and the embedding shares the

same direction of a single code. If $\alpha_{ij} = (1 - 2/c)/\sqrt{n}$, the contribution of the spurious term becomes non-zero, scaled by a factor of $2/c$. This notion of alignment is important to achieve decorrelated and clustered features, as we will see shortly. The key assumptions to the theorem are the orthogonality of $\boldsymbol{W}$, whose codes define a basis for the embedding space and consequently each embedding can be expressed as a linear combination of the dictionary codes, and normalized embeddings, which allow to constrain the possible values of coefficients for this linear combination. It is important to specify that, for the sake of generality, the theorem considers an orthogonal dictionary matrix, which deviates from the specific choice made in Eq. 1. We will elaborate this detail about $\boldsymbol{W}$ in the next subsection.

The theorem has three important consequences that are distilled in the following corollaries:

**Corollary 1** (Perfect alignment). *Given the assumptions in Theorem 1, if $c \to \infty$, then $\forall i \in [n], \exists! j \in [c]$ such that $\alpha_{ij} = \frac{1}{\sqrt{n}}$ is unique and $\boldsymbol{h}_i = \frac{1}{\sqrt{n}}\boldsymbol{w}_j$.*

*Proof.* The result in Theorem 1 states that $\forall i \in [n], \exists! j \in [c]$, the coefficients $\alpha_{ij} \in \{1/\sqrt{n}, (1 - 2/c)/\sqrt{n}\}$. Taking $c \to \infty$, forces all admissible values of $\alpha_{ij}$ to coincide with a unique value $1/\sqrt{n}$. Substituting this result into Eq. 4 completes the proof. $\qquad \square$

This means that the orthogonality of codes, the large size of the dictionary and the normalization of the embeddings are important inductive biases that are sufficient to guarantee perfect alignment to the codes. Indeed, for a large dictionary, each embedding is assigned to only one of the available codes, avoiding spurious terms. We also prove in Appendix F that

**Corollary 2** (Diagonal covariance). *Given Eq. 4 in Theorem 1 and uniform $\boldsymbol{q}$, assume that $\forall i \in [n], \exists! j \in [c]$ such that $\alpha_{ij} = \frac{1}{\sqrt{n}}$, then the covariance of the embeddings is equal to the identity matrix, i.e. $\boldsymbol{H}^T\boldsymbol{H} = \boldsymbol{I}$.*

The assumptions of the corollary can be satisfied by simply choosing a large dictionary along with the other inductive biases discussed for Corollary 1. These biases are sufficient to ensure that the embeddings are decorrelated and span the entire embedding space, thus avoiding dimensional collapse. Moreover, the importance of decorrelated embeddings translates into the property of having both the embedding and representation matrices with full rank. This ensures improved generalization to supervised linear downstream tasks, as shown in Appendix G. Finally, we prove in Appendix H that

**Corollary 3** (Block-diagonal adjacency). *Given Eq. 4 in Theorem 1 and uniform $\boldsymbol{q}$, assume that $\forall i \in [n], \exists! j \in [c]$ such that $\alpha_{ij} = \frac{1}{\sqrt{n}}$, then the adjacency matrix for the*

*embeddings, i.e.* $\boldsymbol{HH}^T$, *is a block-diagonal matrix with blocks of equal size and their size being equal to* $\frac{n}{c}$.

The orthogonality of the codes and the normalization of the embeddings are therefore sufficient conditions to enforce clustered embeddings. The large size of the dictionary ($c \gg 1$) contributes to decreasing the block size of the adjacency matrix, consequently reducing the effect of intra-cluster collapse.

### 3.3. Practical Considerations

So far, the analysis has focussed on the case of $c = f$, demonstrating that by choosing large $c$ (also large $f$) leads to decorrelated and clustered embeddings and the prevention of dimensional and intracluster collapses. However, in practice, we rarely have control over the size of the representation $f$. The typical learning setting of SSL takes a backbone network with a fixed $f$ and uses a projector to train it. Therefore, it is natural to ask whether our previous results hold with the increase of $c$ when $f$ is fixed. At a first glance, the answer to this question is negative. Indeed, note that all previous results rely on the assumption of orthogonal $\boldsymbol{W}$, so that the codes span the whole embedding space and also act as an orthogonal basis. Since $f$ is fixed, we can have only $f$ orthogonal codes. We can go beyond such limitation and provide an affirmative answer to the above question by probabilistically relaxing the notion of orthogonal $\boldsymbol{W}$ and leveraging principles from hyperdimensional computing (HC) (Kanerva, 2009). More concretely, we can choose codes as in Eq. 1, that is $\boldsymbol{W} \in \{-1, 1\}^{f \times c}$ with elements drawn i.i.d. from a Rademacher distribution, to obtain a quasi-orthogonal dictionary matrix. Indeed, we observe that all columns of $\boldsymbol{W}$ have fixed norm, namely $\|\boldsymbol{w}_j\|_2 = \sqrt{f}$, and that the expected cosine similarity between two codes satisfies the following properties:

$$\mathbb{E}_{\boldsymbol{W}}\{\cos(\boldsymbol{w}_j, \boldsymbol{w}_{j'})\} = \begin{cases} 1 & j = j' \\ 0 & j \neq j' \end{cases} \quad \text{and}$$

$$Var_{\boldsymbol{W}}\{\cos(\boldsymbol{w}_j, \boldsymbol{w}_{j'})\} = \frac{1}{f}, \quad \forall j, j' \in [c] \quad (5)$$

Therefore, the codes are orthogonal to each other on average with the variance being inversely proportional to the size of the representation. In other words, $\boldsymbol{W}^T \boldsymbol{W} = f\boldsymbol{I}$ holds on average independently of the choice of $c$ and all assumptions for Theorem 1 and its corollaries are still satisfied. We illustrate the concept in Figure 3. There are several ways to define random code vectors in HC. We chose to use the multiply-add-permute encoding, which leverages a Rademacher distribution. We refer the reader to a recent survey on HC for more details (Kleyko et al., 2022). This is a form of encoding equipped with simple element-wise addition and multiplication operations to perform algebraic compositions. Notably, the exploitation of the composi-

tional properties of HC is beyond the scope of the current work and left to future work.

In Eq. 1 we have a linear and batch normalization layer to ensure that the representations have well-behaved first-order and second-order statistics throughout training, thus speeding up the training convergence. Some examples are provided in Appendix A. Additionally, we replace the L2-normalization activation in the projector with the hyperbolic tangent one, without affecting the results of the theory.[3] The reason for such choice is to facilitate training: Instead of constraining embeddings to lie on a hypersphere and consequently reduce their dimensionality, we constrain them to lie inside a hypercube while preserving their dimensions. We also provide an experimental comparison between the two activations in terms of generalization performance in Appendix B.

## 4. Experiments

The experimental analysis is divided into four main parts. Firstly, we compare *CPLearn* against non-contrastive approaches from the families of feature decorrelation and cluster-based methods on three image datasets, i.e. SVHN (Netzer et al., 2011), CIFAR-10, CIFAR-100 (Krizhevsky et al., 2009). Secondly, we demonstrate the effects of increasing the dictionary size to validate the results in Corollary 2 and 3 and their implication to generalization on downstream tasks, including clustering and linear probe evaluation. Thirdly, we scale the analysis using a larger backbone on CIFAR-10. Finally, we run the analysis on ImageNet-100. We use a ResNet-8 backbone network with $f = 128$ for SVHN and CIFAR10, and with $f = 256$ for CIFAR-100, following the methodology from (Sansone, 2023). The scaled analysis, leverages a ResNet-18 backbone with $f = 512$. For ImageNet-100, we use a standard small ViT with $f = 384$, following the methodology from (Caron et al., 2021). The $\beta$ parameter in Eq. 7 is chosen from the range $\{0.01, 0.05, 0.1, 0.25, 0.5, 1, 2.5, 5, 10\}$, so that both terms in the objective are minimized. More details are available in Appendix I). We use the repository from (da Costa et al., 2022) for SVHN and CIFAR experiments, and the one from (Caron et al., 2021) for ImageNet-100 experiments. Further details are available in Appendices I, L, M and N.

**Generalization on downstream tasks.** We compare *CPLearn* with Barlow Twins (Zbontar et al., 2021), forcing diagonalization of the embedding cross-covariance, SwAV (Caron et al., 2020), using a Sinkhorn-based clustering layer in the projector, Self-Classifier, using a similar loss (Amrani et al., 2022), and GEDI (Sansone & Manhaeve, 2024), using our loss function in conjunction with a multi-

---

[3]At optimality, embeddings are aligned and all results from the corollaries are still valid.

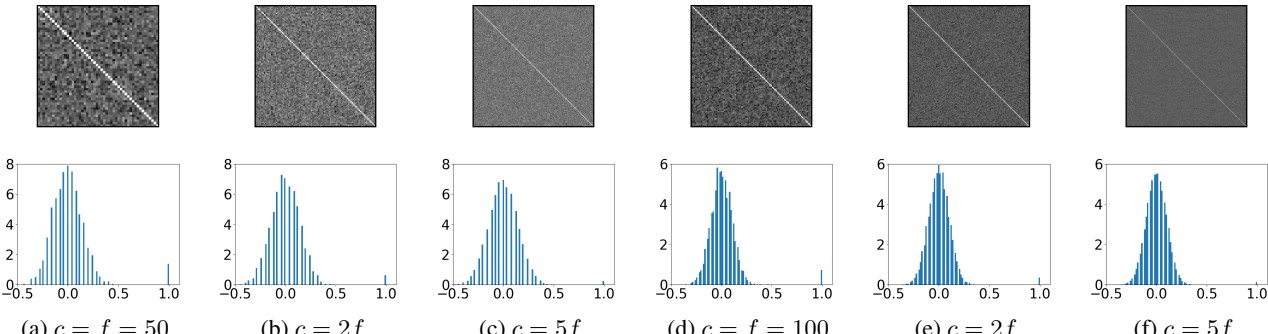

(a) $c = f = 50$     (b) $c = 2f$     (c) $c = 5f$     (d) $c = f = 100$     (e) $c = 2f$     (f) $c = 5f$

Figure 3: **Top:** Illustration of $\boldsymbol{W}^T\boldsymbol{W}$ obtained by randomly sampling $\boldsymbol{W}$. **Bottom:** Normalized histograms of the elements of $\boldsymbol{W}^T\boldsymbol{W}$. Figs. 3a-3c have fixed $f = 50$, whereas Figs. 3d-3f have fixed $f = 100$. $\boldsymbol{W}^T\boldsymbol{W}$ has diagonal values at 1 and random off-diagonal values centered around zero. For larger $c$ and fixed $f$, variance remains constant and quasi-orthogonality is preserved.

Table 1: Test generalization on downstream tasks including clustering and supervised linear probing. Performance are measured in terms of normalized mutual information (NMI), accuracy (Acc.) and are averaged over 5 training runs obtained from random initialization seeds. We test *CPLearn* for undercomplete ($c = 10$), complete ($c = f$) and overcomplete ($c = 16384$) dictionaries. For the Self-Classifier, we test the recommended size for the dictionary ($c = k$, with $k$ being the number of ground truth classes) and the overcomplete case ($c = 16384$).

| Method | Clustering (NMI) | | | Supervised Linear Probing (Acc.) | | |
|---|---|---|---|---|---|---|
| | SVHN | CIFAR-10 | CIFAR-100 | SVHN | CIFAR-10 | CIFAR-100 |
| Barlow | 0.06±0.02 | 0.05±0.01 | 0.10±0.01 | **0.76±0.01** | 0.65±0.00 | 0.28±0.00 |
| SwAV | 0.03±0.01 | 0.29±0.02 | 0.12±0.07 | 0.45±0.03 | 0.56±0.01 | 0.10±0.06 |
| GEDI *no gen* | 0.07±0.02 | 0.33±0.02 | 0.28±0.00 | 0.63±0.02 | **0.66±0.01** | 0.39±0.00 |
| GEDI | 0.07±0.00 | 0.29±0.01 | 0.25±0.01 | 0.58±0.00 | 0.64±0.01 | 0.38±0.00 |
| Self-Classifier ($c = k$) | 0.07±0.02 | 0.28±0.01 | 0.26±0.00 | 0.58±0.01 | 0.59±0.01 | 0.15±0.00 |
| Self-Classifier (16384) | 0.25±0.01 | 0.14±0.10 | 0.17±0.21 | 0.70±0.01 | 0.34±0.18 | 0.16±0.00 |
| *CPLearn* (10) | 0.11±0.01 | 0.28±0.02 | 0.15±0.00 | 0.60±0.02 | 0.59±0.01 | 0.15±0.00 |
| *CPLearn* ($c = f$) | 0.16±0.02 | 0.25±0.01 | 0.33±0.00 | 0.60±0.03 | 0.59±0.01 | 0.18±0.01 |
| *CPLearn* (16384) | **0.29±0.00** | **0.35±0.00** | **0.59±0.00** | **0.75±0.00** | **0.67±0.00** | **0.40±0.00** |

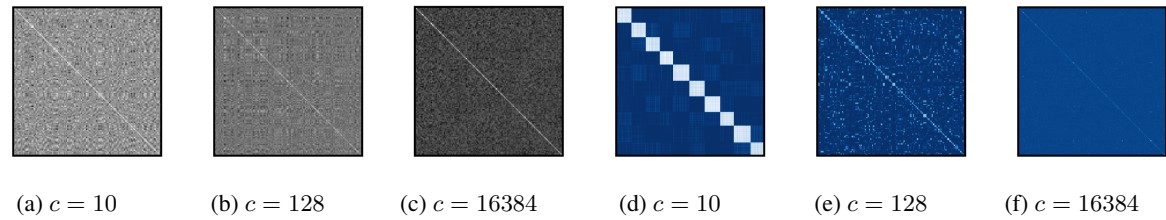

(a) $c = 10$     (b) $c = 128$     (c) $c = 16384$     (d) $c = 10$     (e) $c = 128$     (f) $c = 16384$

Figure 4: Realization of embedding covariance (**left**) and adjacency matrices (**right**) for the whole CIFAR-10 test dataset. Increasing $c$ reduces the value of the off-diagonal elements of the covariance, thus contributing to increase the decorrelation of features (cf. Corollary 2). Moreover, increasing $c$ has the effect to reduce the block sizes of the adjacency matrix (cf. Corollary 3).

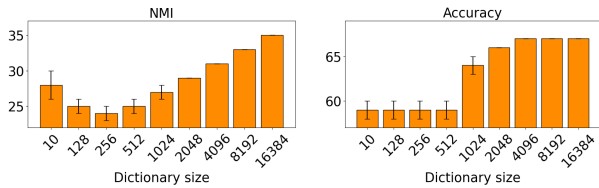

Figure 5: Downstream generalization on CIFAR-10 test dataset, clustering (**left**) and linear evaluation results (**right**).

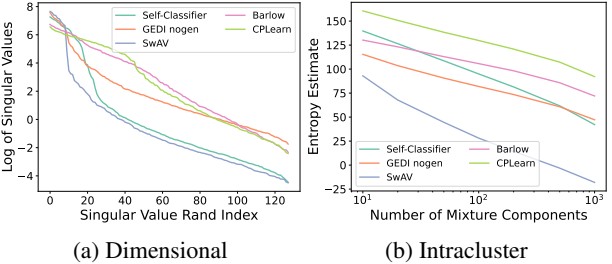

(a) Dimensional        (b) Intracluster

Figure 6: Analysis of dimensional and intracluster collapses. For Self-Classifier, we choose $c = f$.

layer perceptron projector as in Barlow Twins.[4] We test the downstream performance on clustering using normalized mutual information (NMI) computed between the projector predictions and ground truth labels and supervised linear probing on the representations using accuracy (Acc.). In Table 1, we report all results and include three different settings for *CPLearn*, each corresponding to the undercomplete ($c<f$), complete ($c=f$) and overcomplete ($c>f$) dictionary case. We observe that Barlow Twins performs well on linear probe evaluation compared to the other baselines, thanks to the connection between feature decorrelation and generalization (cf. Appendix G), whereas SwAV, Self-Classifier and GEDI perform better on clustering tasks than Barlow Twins. We also observe that *CPLearn* in the undercomplete case performs comparably well to the other cluster-based baselines despite its difference in the design of the projector. However, *CPLearn* is the only method to systematically leverage larger dictionaries to improve both the clustering and classification performance, as predicted by our theory.

**Effects of increasing the dictionary size.** We provide additional insights on the benefits of increasing the size of the dictionary. Specifically, we show some examples of embedding covariances and adjacency matrices computed on CIFAR-10 for different values of $c$ in Fig. 4, thus demonstrating that larger dictionary sizes contribute to implicitly diagonalize the covariance as well as to reduce the block sizes in the adjacency, as predicted by our corollaries. Moreover, we provide a quantitative evaluation on the downstream tasks in Fig. 5, where we observe a monotonic increase in clustering and classification performance with large values of $c$. Similar results hold for other datasets, cf. Appendix J. Interestingly, the monotonic increase in clustering performance suggests that more and more structured information is preserved in the projector.

Additionally, we investigate dimensional collapse following the methodology proposed in previous work (Jing et al., 2022), by computing the singular value distribution of the covariance matrix of the embeddings. We also propose to

---

[4]The work in (Sansone & Manhaeve, 2024) proposes two solutions, one adding a generative term to our objective, named GEDI, and one without, named GEDI *no gen*. Both versions use a standard MLP network for the projector.

study intra-cluster collapse by estimating the entropy of the embedding distribution. This is done by: (i) fitting a Gaussian mixture model with diagonal covariance to the embeddings; (ii) estimating the entropy of the resulting distribution via Monte Carlo sampling using $10k$ samples; and (iii) repeating the analysis for different numbers of mixture components, i.e., $10, 20, 50, 100, 200, 500, 1000$. Higher entropy values indicate a lower degree of intra-cluster collapse. In Appendix K, we (i) qualitatively demonstrate how the loss function helps avoid representation and cluster collapse, and we (ii) quantitatively show how large dictionaries prevent dimensional and intracluster collapse, thus confirming that more structural information is preserved in the projector. Finally, we repeat the analysis of collapses by comparing *CPLearn* to competing methods. In Fig. 6a, we observe that *CPLearn* achieves performance comparable to the best baseline Barlow Twins. This is due to feature decorrelation property induced by our design. All other approaches face dimensional collapse. In Fig.6b, we further demonstrate that *CPLearn* shows increased robustness to intra-cluster collapse compared to its competitors.

| Method | Clustering (NMI) | Linear (Acc.) |
|---|---|---|
| Barlow | 29.1 | **92.2** |
| SwAV | 18.9 | 89.6 |
| GEDI *no gen* | 44.6 | 80.0 |
| Self-Classifier ($c = f$) | 36.9 | 84.8 |
| Self-Classifier (16384) | 33.9 | 64.9 |
| *CPLearn* ($c = f$) | 47.4 | 91.6 |
| *CPLearn* (16384) | **48.2** | 91.3 |

Table 2: Downstream generalization on Resnet-18.

**Analysis with larger backbone.** We perform experiments with a larger backbone, i.e. ResNet-18, trained for 1000 epochs on CIFAR-10 comparing all methods in terms of linear classification and clustering. Further details about hyperparameters are given in Appendix L. Results are shown in Table 2. Overall, the table highlights similar observations to the ones from experiments on ResNet-8, with *CPLearn* achieving comparable performance to Barlow Twins and significantly outperforming all other approaches in terms of clustering. Additional analysis is provided in Appendix L.

Table 3: Test generalization on downstream tasks including clustering and supervised linear probing. Performance are measured in terms of normalized mutual information (NMI), top-1 accuracy (Acc.) Models are trained on ImageNet-100 for 300 epochs. *oom* stands for out-of-memory. Obtaining this table roughly costs 2k euros ($\approx$ 48 GPU hours per simulation * 40 simulations * 1.35 euros per GPU hour on A100 GPUs). In bold, the best performance for each method.

| | Clustering (NMI) | | | | | | Supervised Linear Probing (Acc.) | | | | | |
| | Small Projector | | | Large Projector | | | Small Projector | | | Large Projector | | |
| Method | 128 | 1024 | 2048 | 32768 | 65536 | 131072 | 128 | 1024 | 2048 | 32768 | 65536 | 131072 |
| Barlow | 30.0% | 53.6% | **59.4%** | *oom* | *oom* | *oom* | 66.9% | **77.2%** | **77.2%** | *oom* | *oom* | *oom* |
| SwAV | 28.0% | 47.9% | 51.2% | 60.7% | 60.5% | **62.8%** | 76.6% | 77.7% | **78.0%** | 76.6% | 77.7% | 77.3% |
| DINO | 46.0% | 53.4% | 55.2% | 63.1% | **64.7%** | 64.3% | 71.8% | 73.6% | 73.9% | 75.1% | **76.2%** | 75.8% |
| GEDI *no gen* | 24.5% | 36.3% | **38.8%** | 32.7% | 32.9% | 33.3% | 71.8% | **73.2%** | 72.8% | 72.9% | 72.7% | 72.8% |
| *CPLearn* | 34.1% | 54.3% | 57.9% | 69.3% | 69.6% | **70.9%** | 70.8% | 72.8% | 70.1% | 74.5% | 74.7% | **78.0%** |

**Analysis on ImageNet-100.** We analyze the effect of the dictionary size on ImageNet-100 using a ViT-small backbone and compare *CPLearn* against Barlow Twins (Zbontar et al., 2021), SwAV (Caron et al., 2020), GEDI (Sansone & Manhaeve, 2024) and DINO (Caron et al., 2021). We use the original DINO codebase for the experiments and train all models for 300 epochs. Table 3 summarizes the results in terms of clustering and linear probing evaluation. While Barlow Twins and SwAV achieve good generalization on the linear probing task, they underperform compared to DINO in terms of clustering performance, demonstrating that no method is capable to tackle both tasks well. In contrast, *CPLearn* is able to make effective use of a large dictionary size to achieve good generalization on both clustering and linear probing tasks. This further demonstrates the value of the theoretical guarantees of *CPLearn*, translating into a simplified and principled design of the projector and loss function compared to DINO (such as avoiding the use of asymmetric operations like stop gradient, centering operation for the teacher network, use of different temperature parameters for student and teacher networks and exponential moving average update of the teacher parameters), while ensuring good properties like decorrelated and clustered features.

## 5. Conclusions

We have distilled the essential principles of non-contrastive self-supervised learning into the design of a projector and loss function that prevent known failure modes, including representation, cluster, dimensional, and intracluster collapses. This approach enables robust training of a backbone network, achieving representations that are both decorrelated and clustered. We have also demonstrated that the resulting solutions improve generalization to supervised downstream tasks when large dictionaries are used in the projector. A future direction is to improve scalability when storing and using large dictionaries. We could leverage the bipolar nature of the codes, instead of treating them as floating-point vectors, to significantly enhance efficiency. In

the future, we plan to leverage the connection with hyperdimensional computing and leverage its algebraic properties for tackling learning and reasoning tasks.

## Impact Statement

While the results of this work are foundational in nature, several positive impacts can be anticipated. Specifically, this work (i) enhances the robustness of non-contrastive self-supervised learning against training failures and (ii) simplifies the training design, taking a meaningful step toward more trustworthy representation learning strategies and democratizing access to their solutions.

## Author Contributions

E.S. had the original idea, developed the theory, wrote the code, ran the experiments, wrote the paper. T. L. implemented the initial distributed code and helped with scaling results on ResNet-18 and ImageNet. T. T. provided the computational resources to run the experiments in Table 1, provided feedback and support in proof-reading the article.

## Acknowledgments

This work received funding from the European Research Council (ERC) under the European Union's Horizon 2020 research and innovation programme (grant agreement n° 101021347) and under the Horizon Europe research and innovation programme (MSCA-GF grant agreement n° 101149800). The computational resources and services used in this work were provided by the VSC (Flemish Supercomputer Center), funded by the Research Foundation Flanders (FWO) and the Flemish Government – department EWI. Moreover, we acknowledge Euro CC Belgium for awarding this project (n° 465001817) access to the LUMI supercomputer, owned by the EuroHPC Joint Undertaking and hosted by CSC (Finland) and the LUMI consortium. Any opinions, findings, and conclusions or recommendations expressed in this material are those of the authors and do not necessarily reflect the views of our sponsors.

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

# Appendix

## A. Training Statistics

We provide some examples demonstrating the effectiveness of introducing linear and batch normalization layers in the projector in Fig. 7. This contributes to prevent excessing increase of mean or variance. An alternative and promising approach to the linear and batch normalization layers is to penalize the norm of the representations directly in the objective. We leave this to future investigation.

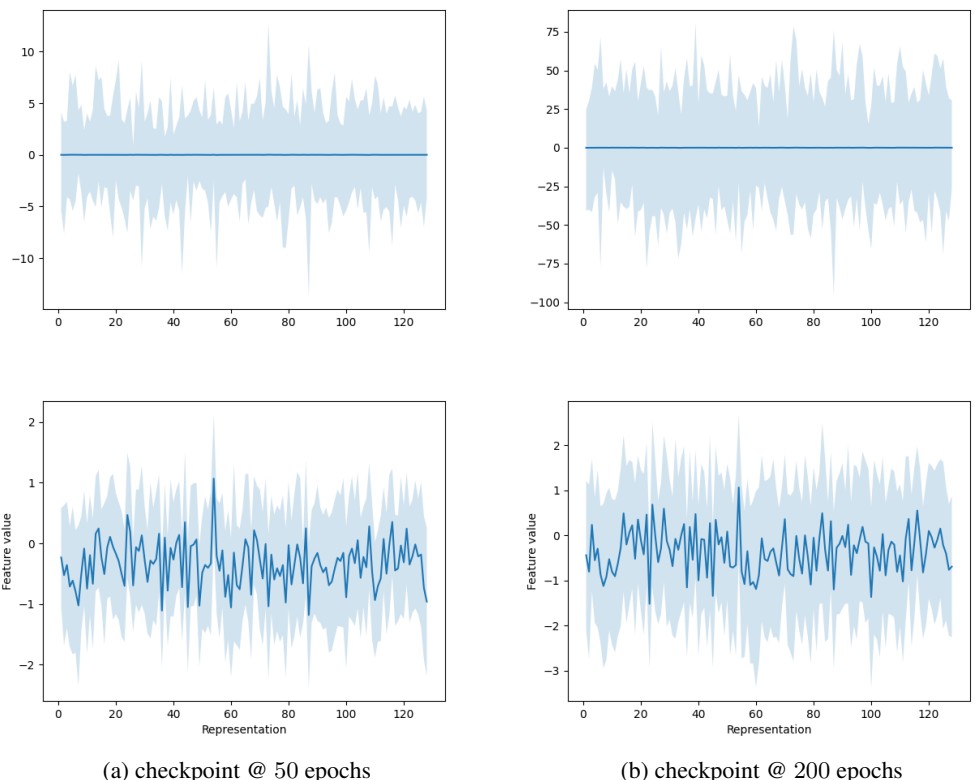

(a) checkpoint @ 50 epochs         (b) checkpoint @ 200 epochs

Figure 7: Example of mean and standard deviation statistics for the representation features obtained by the backbone network trained on SVHN data. Statistics are computed for a batch of size of 100 samples. Results corresponds to checkpoints for the projector ($c = 4096$) without (**top**) and with (**bottom**) linear and batch normalization layers. Linear and batch normalization layers contribute to stabilize the training by avoiding mean or variance increase.

## B. Comparison of Different Activations for the Projector

We provide a comparison between activation functions on SVHN, CIFAR-10 and CIFAR-100 in Table 4. For both activations, we use $\beta$ according to Table 7.

## C. Discussion on Finite Capacity

It is important to mention that the global minima for the *CPLearn* objective might not be reached when using a backbone network of finite and small capacity. In this case, the avoidance of representation and cluster collapses can still be guaranteed when the invariance and the matching prior losses are both minimized. Indeed, we observe that for representation collapse $p_{ij} = p_j$ for all $i \in [n], j \in [c]$ (i.e. the outputs of the overall network are constant with respect to their inputs) and that the corresponding mimimum value of the objective is given by the following formula

$$\mathcal{L}_{CPLearn}(\mathcal{D}) = \beta H(\boldsymbol{p}) + CE(\boldsymbol{q}, \boldsymbol{p})$$

Table 4: Test generalization on downstream tasks including clustering and supervised linear probing. Performance are measured in terms of normalized mutual information (NMI), accuracy (Acc.) and are averaged over 5 training runs obtained from random initialization seeds. We test *CPLearn* using L2-Norm and Tanh activations.

| $c$ | Activation | Clustering (NMI) | | | Supervised Linear Probing (Acc.) | | |
|---|---|---|---|---|---|---|---|
| | | SVHN | CIFAR-10 | CIFAR-100 | SVHN | CIFAR-10 | CIFAR-100 |
| 10 | L2-Norm | 0.14±0.02 | 0.30±0.01 | 0.16±0.01 | 0.64±0.02 | 0.60±0.01 | 0.15±0.01 |
| | Tanh | 0.11±0.01 | 0.28±0.02 | 0.15±0.00 | 0.60±0.02 | 0.59±0.01 | 0.15±0.00 |
| $f$ | L2-Norm | 0.16±0.00 | 0.26±0.00 | 0.33±0.01 | 0.59±0.01 | 0.60±0.01 | 0.20±0.01 |
| | Tanh | 0.16±0.02 | 0.25±0.01 | 0.33±0.00 | 0.60±0.03 | 0.59±0.01 | 0.18±0.01 |
| 16384 | L2-Norm | **0.31±0.00** | **0.35±0.00** | 0.58±0.00 | **0.78±0.01** | **0.68±0.00** | **0.41±0.00** |
| | Tanh | 0.29±0.00 | **0.35±0.00** | **0.59±0.00** | 0.75±0.00 | **0.68±0.00** | 0.40±0.00 |

where the first addend arises from the invariance loss, whereas the second one arises from the matching prior one. Notably, the two terms cannot be minimized at the same time due to their competitive nature. For instance, in the case of uniform $q$, the solution of $p = q$ is a minimum for the matching prior loss but not for the invariance one (this is actually a saddle point, as corresponding to the maximum for the entropy term in the above equation).

Cluster collapse occurs whenever $\exists j, k \neq j \in [c]$ such that for all $i \in [n]$, $p_{ij} \leq p_{ik}$. The minimization of the invariance loss forces the whole network to make low entropy predictions, whereas the minimization of the matching prior loss forces to distribute these predictions across all codes according to $q$. Hence, when both losses are minimized cluster collapse is avoided.

## D. Minima of the *CPLearn* Loss

*Proof.* We recall here the loss

$$\mathcal{L}_{CPLearn}(\mathcal{D}) = -\frac{\beta}{n} \sum_{i=1}^{n} \sum_{j=1}^{c} p_{ij} \log p'_{ij} - \sum_{j=1}^{c} q_j \log \frac{1}{n} \sum_{i=1}^{n} p_{ij}$$

and prove all optimality conditions. Before doing that, we observe that the loss is convex w.r.t. $\mathbf{P}$ when $\mathbf{P}'$ is fixed, as the first addend is a sum of linear terms, whereas the second addend is a sum of convex terms. Similarly, we observe that convexity holds w.r.t. $\mathbf{P}'$ when $\mathbf{P}$ is fixed by exploiting the same reasoning. However, it is important to mention that the loss is not convex globally. This can be shown firstly by computing the Hessian of the first addend w.r.t. both $\mathbf{P}$ and $\mathbf{P}'$ and secondly by observing that the Hessian is not positive semi-definite (we skip the tedious calculation of the Hessian).

**Invariance.** We observe that $\mathbf{P}'$ appears only in the first addend of $\mathcal{L}_{CPLearn}$ and that this addend can be equivalently rewritten in the following way:

$$-\frac{\beta}{n} \sum_{i=1}^{n} \sum_{j=1}^{c} p_{ij} \log p'_{ij} = -\frac{\beta}{n} \sum_{i=1}^{n} \sum_{j=1}^{c} p_{ij} \log p_{ij} - \frac{\beta}{n} \sum_{i=1}^{n} \sum_{j=1}^{c} p_{ij} \log \frac{p'_{ij}}{p_{ij}}$$

$$= \frac{\beta}{n} \sum_{i=1}^{n} H(\mathbf{p}_i) + \frac{\beta}{n} \sum_{i=1}^{n} KL(\mathbf{p}_i \| \mathbf{p}'_i) \qquad (6)$$

where $H(.), KL(.)$ are the entropy and Kullback-Leibler divergence, respectively. Therefore minimizing $\mathcal{L}_{CPLearn}$ w.r.t. $\mathbf{P}'$ is equivalent to minimizing Eq. 6. The solution is given by $\mathbf{p}_i = \mathbf{p}'_i, \forall i \in [n]$, thus proving the invariance condition.

**Extrema.** We first leverage the invariance condition, $\mathbf{p}_i = \mathbf{p}'_i, \forall i \in [n]$, and rewrite $\mathcal{L}_{CPLearn}$ accordingly:

$$\mathcal{L}_{CPLearn}(\mathcal{D}) = -\frac{\beta}{n} \sum_{i=1}^{n} \sum_{j=1}^{c} p_{ij} \log p_{ij} - \sum_{j=1}^{c} q_j \log \frac{1}{n} \sum_{i=1}^{n} p_{ij} \qquad (7)$$

We observe that the loss in Eq. 7 is convex w.r.t. $\mathbf{P}$. Therefore, we can obtain its optimality conditions, by deriving the closed-form solutions for the minima of the second addend in Eq. 7, and then constraining the optimization of the first addend with these solutions and deriving the corresponding minima.

Let's start by considering the following constrained convex minimization problem, obtained from the first addend in Eq. 7, with $n, \beta$ being dropped as being constant for the optimization:

$$\min_{\mathbf{P}} -\sum_{i=1}^{n}\sum_{j=1}^{c} p_{ij} \log p_{ij}$$

$$\text{s.t.} \quad \sum_{j=1}^{c} p_{ij} = 1, \quad \forall i \in [n]$$

$$\epsilon \leq p_{ij} \leq 1 - \epsilon(c-1), \quad \forall i \in [n], j \in [c], \tag{8}$$

and the corresponding Lagrangian with multipliers $\mathbf{\Lambda}, \mathbf{\Delta} \in \mathbb{R}_{+}^{n \times c}, \boldsymbol{\nu} \in \mathbb{R}^{n}$ is:

$$\mathcal{L}_1(\mathbf{P}; \mathbf{\Lambda}, \mathbf{\Delta}, \mathbf{\Omega}, \boldsymbol{\nu}) \equiv -\sum_{i=1}^{n}\sum_{j=1}^{c} p_{ij} \log p_{ij} + \sum_{i=1}^{n} \nu_i \left( \sum_{j=1}^{c} p_{ij} - 1 \right) +$$

$$+ \sum_{i=1}^{n}\sum_{j=1}^{c} [\lambda_{ij}(\epsilon - p_{ij}) + \delta_{ij}(p_{ij} - 1 + \epsilon(c-1))] \tag{9}$$

We observe that the Lagrangian is constructed so as to satisfy the following relation

$$-\sum_{i=1}^{n}\sum_{j=1}^{c} p_{ij} \log p_{ij} \geq \mathcal{L}_1(\mathbf{P}; \mathbf{\Lambda}, \mathbf{\Delta}, \mathbf{\Omega}, \boldsymbol{\nu}) \tag{10}$$

Let's maximize $\mathcal{L}_1$ w.r.t. $\mathbf{P}$ by setting $\nabla_{p_{ij}} \mathcal{L}_1 = 0$. This leads to the following closed-form expression:

$$p_{ij}^* = e^{-1 - \lambda_{ij} + \nu_i + \delta_{ij}} \quad \forall i \in [n], j \in [c] \tag{11}$$

By evaluating $\mathcal{L}_1$ at the solutions in Eq. 11, we obtain the Lagrange dual function

$$\mathcal{L}_1(\mathbf{P}^*; \mathbf{\Lambda}, \mathbf{\Delta}, \mathbf{\Omega}, \boldsymbol{\nu}) = n + \sum_{i=1}^{n} \left\{ -\nu_i + \sum_{j=1}^{c} [\lambda_{ij}\epsilon - \delta_{ij}(1 - \epsilon(c-1))] \right\} \tag{12}$$

The Lagrange multipliers in Eq. 12 depend on the values of $\mathbf{P}^*$ through the Karush-Kuhn-Tucker (KKT) conditions. We distinguish two main cases for $\mathbf{P}^*$, each leading to different evaluation of the Lagrange dual function:

- *Case 1.* When all probability values touch their extrema, such as

$$\forall i \in [n], \exists! j \in [c], \forall k \in [c] \text{ with } k \neq j \text{ s.t. } p_{ij}^* = 1 - \epsilon(c-1) \text{ and } p_{ik}^* = \epsilon$$

By the KKT conditions (i.e. complementary slackness), we have that $\lambda_{ij} = 0$ and $\delta_{ik} = 0$, whereas $\lambda_{ik} \geq 0, \delta_{ij} \geq 0$. By substituting these conditions in Eq. 12, we obtain that

$$\mathcal{L}_1(\mathbf{P}^*; \mathbf{\Lambda}, \mathbf{\Delta}, \mathbf{\Omega}, \boldsymbol{\nu})|_{\{\lambda_{ij}=\delta_{ik}=0\}} = n + \sum_{i=1}^{n} \left\{ -\nu_i - \delta_{ij}(1 - \epsilon(c-1)) + \sum_{k \neq j} \lambda_{ik}\epsilon \right\} \tag{13}$$

By taking into account also Eq. 11, we have that $\forall i \in [n], \exists! j \in [c], \forall k \in [c]$

$$\delta_{ij} = 1 - \nu_i + \log(1 - \epsilon(c-1)) \text{ and } \lambda_{ik} = -1 + \nu_i - \log \epsilon \tag{14}$$

And by substituting Eq. 14 into Eq. 13, we obtain that

$$\mathcal{L}_1(\mathbf{P}^*; \mathbf{\Lambda}, \mathbf{\Delta}, \mathbf{\Omega}, \boldsymbol{\nu})|_{\{\lambda_{ij} = \delta_{ik} = 0\} \text{ and Eq. 14}} = -n(1 - \epsilon(c-1)) \log(1 - \epsilon(c-1)) - \\ - n\epsilon(c-1) \log \epsilon \tag{15}$$

- *Case 2.* When all probability values never touch the highest extrema, such as

$$\forall i \in [n], j \in [c], \text{ s.t. } p_{ij}^* < 1 - \epsilon(c-1)$$

By KKT conditions, we have that $\delta_{ij} = 0$. By substituting these conditions in Eq. 12, we obtain that

$$\mathcal{L}_1(\mathbf{P}^*; \mathbf{\Lambda}, \mathbf{\Delta}, \mathbf{\Omega}, \boldsymbol{\nu})|_{\{\delta_{ij} = 0\}} = n + \sum_{i=1}^{n} \left\{ -\nu_i + \sum_{j=1}^{c} \lambda_{ij} \epsilon \right\} \tag{16}$$

which always satisfies the inequality

$$\mathcal{L}_1(\mathbf{P}^*; \mathbf{\Lambda}, \mathbf{\Delta}, \mathbf{\Omega}, \boldsymbol{\nu})|_{\{\delta_{ij} = 0\}} \geq \mathcal{L}_1(\mathbf{P}^*; \mathbf{\Lambda}, \mathbf{\Delta}, \mathbf{\Omega}, \boldsymbol{\nu})|_{\{\lambda_{ij} = \delta_{ik} = 0\}} \tag{17}$$

and therefore also

$$\mathcal{L}_1(\mathbf{P}^*; \mathbf{\Lambda}, \mathbf{\Delta}, \mathbf{\Omega}, \boldsymbol{\nu})|_{\{\delta_{ij} = 0\}} \geq \mathcal{L}_1(\mathbf{P}^*; \mathbf{\Lambda}, \mathbf{\Delta}, \mathbf{\Omega}, \boldsymbol{\nu})|_{\{\lambda_{ij} = \delta_{ik} = 0\} \text{ and Eq. 14}} \tag{18}$$

Finally, we observe that the objective of the optimization problem of Eq. 8 evaluated at the solutions of *Case 1* is

$$-\sum_{i=1}^{n} \sum_{j=1}^{c} p_{ij} \log p_{ij} = \mathcal{L}_1(\mathbf{P}^*; \mathbf{\Lambda}, \mathbf{\Delta}, \mathbf{\Omega}, \boldsymbol{\nu})|_{\{\lambda_{ij} = \delta_{ik} = 0\} \text{ and Eq. 14}} \tag{19}$$

And by leveraging also the result in Eq. 18, we can state that the solutions of *Case 1* are the global minima of the objective in Eq. 8. Thus concluding the proof for the extrema condition.

**Matched prior.** We consider the minimization of the second addend in Eq. 7 subject to the extrema condition

$$\min_{\mathbf{P}} -\sum_{j=1}^{c} q_j \log \frac{1}{n} \sum_{i=1}^{n} p_{ij}$$
$$\text{s.t.} \quad \sum_{j=1}^{c} p_{ij} = 1, \quad \forall i \in [n]$$
$$p_{ij} \in \{\epsilon, 1 - \epsilon(c-1)\}, \quad \forall i \in [n], j \in [c], \tag{20}$$

Let's define $\tilde{p}_j \equiv \frac{1}{n} \sum_{i=1}^{n} p_{ij}$ for all $j \in [c]$ and observe that $\sum_{j=1}^{c} \tilde{p}_j = 1$ and $\epsilon \leq \tilde{p}_j \leq 1 - \epsilon(c-1)$. Therefore, we can rewrite the problem in Eq. 20 equivalently

$$\min_{\mathbf{P}} -\sum_{j=1}^{c} q_j \log \tilde{p}_j$$
$$\text{s.t.} \quad \sum_{j=1}^{c} \tilde{p}_j = 1,$$
$$\epsilon \leq \tilde{p}_j \leq 1 - \epsilon(c-1), \quad \forall j \in [c], \tag{21}$$

Now, we observe that the optimization objective satisfies the following equality

$$-\sum_{j=1}^{c} q_j \log \tilde{p}_j = H(\boldsymbol{q}) + KL(\mathbf{q}\|\tilde{\boldsymbol{p}}) \tag{22}$$

The minimum for Eq. 22 is obtained at $\mathbf{q} = \tilde{\boldsymbol{p}}$ and this solution satisfies the constraints in Eq. 21 because $\epsilon \leq q_j \leq 1-\epsilon(c-1)$ for all $j \in [c]$ (indeed we can always choose $\epsilon$ to satisfy the inequality), thus being the global optimum. In other words, we have that $\frac{1}{n}\sum_{i=1}^{n} p_{ij} = q_j$ for all $j \in [c]$.

Finally, recall that $I_{max}(j) \equiv \{i \in [n] : p_{ij} = 1 - \epsilon(c-1)\}, \forall j \in [c]$, which identifies all elements having the highest possible value of probability in $\boldsymbol{P}$. We observe that

$$\begin{aligned}
\sum_{i=1}^{n} p_{ij} &= \sum_{i \in I_{max}(j)} p_{ij} + \sum_{i \notin I_{max}(j)} p_{ij} \\
&= \sum_{i \in I_{max}(j)} (1 - \epsilon(c-1)) + \sum_{i \notin I_{max}(j)} \epsilon \quad \text{(by Extrema condition)} \\
&= |I_{max}(j)|(1 - c\epsilon) + n\epsilon
\end{aligned}$$

By the condition $\frac{1}{n}\sum_{i=1}^{n} p_{ij} = q_j$ and the above relation we have that

$$|I_{max}(j)|(1 - c\epsilon) + n\epsilon = nq_j, \quad \forall j \in [c]$$

or equivalently that

$$|I_{max}(j)| = \left(\frac{q_j - \epsilon}{1 - c\epsilon}\right) n \tag{23}$$

Now, for the case of uniform prior, Eq. 23 becomes

$$q_j = \frac{1}{c} \implies |I_{max}(j)| = \frac{n}{c}, \quad \forall j \in [c] \tag{24}$$

This concludes the proof for the matching prior condition.

Finally the global minimum value of the *CPLearn* objective can be obtained by dividing Eq. 15 by $n$ and adding the entropy term (as for the result obtained by the matched prior condition). This concludes the proof of the Lemma. □

## E. Embedding Theorem

*Proof.* Recall the extrema condition from Lemma 1, that is

$$\forall i \in [n], \exists! j \in [c], \forall k \in [c] \text{ with } k \neq j \text{ s.t. } p_{ij} = 1 - \epsilon(c-1) \text{ and } p_{ij} = \epsilon$$

Moreover, due to orthogonality of $\boldsymbol{W}$ we can express the *Span* condition, i.e. $\boldsymbol{h}_i = \sum_{j'=1}^{c} \alpha_{ij'}\boldsymbol{w}_{j'}$ for all $i \in [n]$ with $\alpha_{ij} \in \mathbb{R}$, This fact leads us to the following equation

$$p_{ij} = \frac{e^{\boldsymbol{w}_j^T \boldsymbol{h}_i/\tau}}{\sum_{j''=1}^{c} e^{\boldsymbol{w}_{j''}^T \boldsymbol{h}_i/\tau}} \underbrace{=}_{Span} \frac{e^{\alpha_{ij} f/\tau}}{\sum_{j'=1}^{c} e^{\alpha_{ij'} f/\tau}} \quad \forall i \in [n], j \in [c] \tag{25}$$

Combining the extrema condition with Eq. 25 gives us a system of equations for each $i \in [n]$

$$\begin{cases}
\frac{e^{\alpha_{ij} f/\tau}}{\sum_{j'=1}^{c} e^{\alpha_{ij'} f/\tau}} = 1 - \epsilon(c-1) & \\
\frac{e^{\alpha_{ik} f/\tau}}{\sum_{j'=1}^{c} e^{\alpha_{ij'} f/\tau}} = \epsilon & \forall k \neq j
\end{cases}$$

By taking the logarithm on both sides of the two equations and resolving the above system, the solution is equal to

$$\alpha_{ik} = \alpha_{ij} - \frac{\tau}{f} \log\left(\frac{1 - \epsilon(c-1)}{\epsilon}\right)$$

$$= \alpha_{ij} - \frac{1}{\sqrt{n}} \quad \forall k \neq j \tag{26}$$

where the last equality holds due to the choice $\tau = f/(\sqrt{n}\log((1-\epsilon(c-1))/\epsilon))$. Using Eq. 26 in the *Span* condition gives us the following result

$$\forall i \in [n], \exists! j \in [c] \text{ s.t. } \boldsymbol{h}_i = \alpha_{ij}\boldsymbol{w}_j + \left(\alpha_{ij} - \frac{1}{\sqrt{n}}\right)\sum_{k \neq j} \boldsymbol{w}_k \tag{27}$$

Note that the $\alpha_{ij}$ could potentially take any value in $\alpha_{ij} \in \mathbb{R}$. This is not allowed as embeddings are normalized by design choice (cf. Eq. 1). Indeed, the norm of the embeddings can be rewritten to exploit Eq. 27

$$\|\boldsymbol{h}_i\|_2^2 = \boldsymbol{h}_i^T \boldsymbol{h}_i$$

$$\underbrace{=}_{\text{Eq. 27}} cf\alpha_{ij}^2 - \frac{2(c-1)f}{\sqrt{n}}\alpha_{ij} + \frac{f}{n}(c-1) \tag{28}$$

and by equating Eq. 28 to the fact that embeddings are normalized $\|\boldsymbol{h}_i\|_2^2 = \frac{f}{n}$ for all $i \in [n]$ we obtain the following quadratic equation

$$\alpha_{ij}^2 - \frac{2(c-1)f}{\sqrt{n}}\alpha_{ij} + \frac{f}{n}(c-1) - \frac{f}{n} = 0$$

whose solutions are given by

$$\alpha_{ij} = \left\{ \begin{array}{l} \frac{1}{\sqrt{n}} \\ \left(1 - \frac{2}{c}\right)\frac{1}{\sqrt{n}} \end{array} \right.$$

This concludes the proof. □

## F. Diagonal Covariance

*Proof.* Recall from Theorem 1 that

$$\forall i \in [n], \exists! j \in [c] \text{ s.t. } \boldsymbol{h}_i = \alpha_{ij}\boldsymbol{w}_j + \left(\alpha_{ij} - \frac{1}{\sqrt{n}}\right)\sum_{k \neq j} \boldsymbol{w}_k$$

By assumption $\alpha_{ij} = \frac{1}{\sqrt{n}}$ and therefore

$$\forall i \in [n], \exists! j \in [c] \text{ s.t. } \boldsymbol{h}_i = \frac{1}{\sqrt{n}}\boldsymbol{w}_j \tag{29}$$

meaning that the rows of $\boldsymbol{H}$ are equal up to a constant to the codes in the dictionary and that they span the same space of the columns of $\boldsymbol{W}$, namely the whole embedding space. We can therefore express $\boldsymbol{H}$ as linear combination of $\boldsymbol{W}$.

Without loss of generality, we can always define $\boldsymbol{H}$ so as to ensure that nearby rows are associated to the same codes in the dictionary. Therefore, by combining this with Eq. 29 we have that

$$\boldsymbol{H} = \boldsymbol{A}^T \boldsymbol{W}^T$$

with

$$A = \begin{pmatrix} \frac{1}{\sqrt{n}}\mathbf{1}_{n/c}^T & \mathbf{0} & \cdots & \mathbf{0} \\ \mathbf{0} & \frac{1}{\sqrt{n}}\mathbf{1}_{n/c}^T & \cdots & \mathbf{0} \\ \vdots & \vdots & \ddots & \vdots \\ \mathbf{0} & \mathbf{0} & \cdots & \frac{1}{\sqrt{n}}\mathbf{1}_{n/c}^T \end{pmatrix} \in \mathbb{R}^{c \times n}$$

where $\mathbf{1}_{n/c}$ is a vector containing $n/c$ ones (whose size follows due to the assumption on uniformity of $q$). Importantly, matrix $A$ satisfies the following property

$$AA^T = \frac{1}{c}I \tag{30}$$

Therefore, we have that

$$H^T H = WAA^T W^T$$
$$\underbrace{=}_{\text{Eq. 30}} \frac{1}{c}WW^T$$
$$= I$$

where the last equality simply follows by the orthogonality condition $W^T W = fI$ and the fact that $W$ is a square matrix ($c = f$). Indeed, we have that

$$W^T W = fI$$
$$WW^T W = fIW$$
$$(WW^T)W = (fI)W$$
$$WW^T = fI$$

thus concluding the proof. $\qquad\square$

## G. Generalization to Supervised Linear Downstream Task

We first observe that by the results of Theorem 1 and the uniformity of $q$, $H$ has full rank. Moreover, considering that $H$ is a function of $Z$ through the first layer of the projector in Eq. 1, $Z$ must be also full rank. As a consequence,

$$Z^T Z \text{ has full rank.} \quad \textit{(Full Rank Property)} \tag{31}$$

Now, we recall an existing result for generalization to supervised downstream tasks from (Shwartz-Ziv et al., 2023) (Section 6.1) and demonstrate that the *Full Rank Property* reduces the generalization error.

Indeed, consider a classification problem with $r$ classes. Given an unlabeled dataset $\mathcal{D}$, used for training *CPLearn*, with the corresponding unknown ground truth labels $Y_\mathcal{D} \in \mathbb{R}^{n \times r}$ and a supervised dataset $\mathcal{S} = \{(\boldsymbol{x}_i, \boldsymbol{y}_i)\}_{i=1}^m$, with $\boldsymbol{y}_i$ being the rows of the label matrix $Y_\mathcal{S} \in \mathbb{R}^{m \times r}$, define $Z \in \mathbb{R}^{n \times f}$ and $\bar{Z} \in \mathbb{R}^{m \times f}$ the representations obtained by feeding datasets $\mathcal{D}$ and $\mathcal{S}$, respectively, through the backbone network $g$. Moreover, define

$$P_\mathcal{D} \equiv I - Z(Z^T Z)^\dagger Z^T$$
$$P_\mathcal{S} \equiv I - \bar{Z}(\bar{Z}^T \bar{Z})^\dagger \bar{Z}^T$$

where symbol $\cdot^\dagger$ denotes the pseudo-inverse. Now, suppose we train a linear classifier with parameters $U \in \mathbb{R}^{f \times r}$ on the latent representations obtained from dataset $\mathcal{S}$ through the following supervised loss

$$\ell_{\boldsymbol{x},\boldsymbol{y}}(U) \equiv \|g(\boldsymbol{x})U - \boldsymbol{y}\|_2^2 + \|U\|_F$$

Then, we can state the following theorem

**Th. 1** (restated from (Shwartz-Ziv et al., 2023)). $\forall \delta > 0$ *with probability at least* $1 - \delta$, *we have that*

$$\mathbb{E}_{\boldsymbol{x},\boldsymbol{y}}\{\ell_{\boldsymbol{x},\boldsymbol{y}}(\boldsymbol{U})\} \leq \frac{1}{n}\sum_{i=1}^{n}\|g(\boldsymbol{x}_i) - g(\boldsymbol{x}_i')\|_2 + \frac{2}{m}\mathbb{E}_{\mathcal{D},\boldsymbol{\xi}}\left\{\sup_{g}\sum_{i=1}^{n}\xi_i\|g(\boldsymbol{x}_i) - g(\boldsymbol{x}_i')\|_2\right\} +$$

$$+ \frac{2}{\sqrt{n}}\|\boldsymbol{P}_{\mathcal{D}}\boldsymbol{Y}_{\mathcal{D}}\|_F + \frac{1}{\sqrt{m}}\|\boldsymbol{P}_{\mathcal{S}}\boldsymbol{Y}_{\mathcal{S}}\|_F + const(n,m) \tag{32}$$

*where $\boldsymbol{\xi}$ is a vector of i.i.d. Rademacher random variables.*

Therefore, the expected supervised loss in Eq. 32 can be reduced by minimizing its upper bound. Note that the first addend in Eq. 32 is minimized by the *CPLearn* loss, whereas the second addend is also statistically minimized when $n$ is large. The third addend refers to the contribution term for the classification on the unlabeled data. While ground truth $\boldsymbol{Y}_{\mathcal{D}}$ is unknown, this addend can be minimized by exploiting the following relation

$$\|\boldsymbol{P}_{\mathcal{D}}\boldsymbol{Y}_{\mathcal{D}}\|_F \leq \|\boldsymbol{P}_{\mathcal{D}}\|_F\|\boldsymbol{Y}_{\mathcal{D}}\|_F$$

Indeed, note that in order to minimize the left-hand side of the inequality, it suffices to minimize the term $\|\boldsymbol{P}_{\mathcal{D}}\|_F$, which occurs when $\boldsymbol{Z}^T\boldsymbol{Z}$ has maximum rank. This is our case due to the *Full Rank Property*. Finally, by the same argument used for the third term in Eq. 32, we can minimize the fourth one by having $\bar{\boldsymbol{Z}}^T\bar{\boldsymbol{Z}}$ with maximum rank. This condition holds because $\boldsymbol{Z}^T\boldsymbol{Z}$ and $\bar{\boldsymbol{Z}}^T\bar{\boldsymbol{Z}}$ concentrate to each other by concentration inequalitites (cf. (Shwartz-Ziv et al., 2023) for more details).

To summarize, minimizing the *CPLearn* loss ensures that we reduce the invariance of representations to data augmentations and increase the rank of the representation covariance. This leads to a decrease of the generalization error as from the result of Theorem 1.

## H. Block-Diagonal Adjacency

*Proof.* The proof follows step by step the one for the diagonal covariance except for the fact that

$$\boldsymbol{H}\boldsymbol{H}^T = \boldsymbol{A}^T\boldsymbol{W}^T\boldsymbol{W}\boldsymbol{A} \underbrace{=}_{\boldsymbol{W}^T\boldsymbol{W} = f\boldsymbol{I}} f\boldsymbol{A}^T\boldsymbol{A} = f\boldsymbol{B}_{\boldsymbol{A}}$$

where

$$\boldsymbol{B}_{\boldsymbol{A}} \equiv \boldsymbol{A}^T\boldsymbol{A} = \begin{pmatrix} \boldsymbol{1}_{\frac{n}{c}\times\frac{n}{c}} & \boldsymbol{0} & \cdots & \boldsymbol{0} \\ \boldsymbol{0} & \frac{1}{n}\boldsymbol{1}_{\frac{n}{c}\times\frac{n}{c}} & \cdots & \boldsymbol{0} \\ \vdots & \vdots & \ddots & \vdots \\ \boldsymbol{0} & \boldsymbol{0} & \cdots & \frac{1}{n}\boldsymbol{1}_{\frac{n}{c}\times\frac{n}{c}} \end{pmatrix} \in \mathbb{R}^{n\times n}$$

and $\boldsymbol{1}_{\frac{n}{c}\times\frac{n}{c}}$ is a matrix of ones. This concludes the proof. $\square$

## I. Experimental Details on SVHN, CIFAR10 and CIFAR100

**Training.** We used a ResNet-8 (details are provided in Table 5. We consider the hyperparameters in Table 6 for training. Beta is chosen to ensure both losses are minimized, cf. Table 7.

**Evaluation.** For linear probe evaluation, we followed standard practice by removing the projector head and train a linear classifier on the backbone representation. We train the classifier with Adam optimizer for 100 epochs and learning rate equal to $1e - 2$.

## J. Additional Results on Dictionary Size

We provide additional visualization results for the covariance and adjacency matrices on SVHN and CIFAR-10, cf. Figs. 8, 9. Moreover, we add the analysis of generalization on downstream tasks on SVHN and CIFAR-100 varying the size of the dictionary in Figs 10, 11.

## K. Qualitative and Quantitative Analysis of Collapses

In Fig. 12, we provide some qualitative evidence on the avoidance of representation and cluster collapses. Indeed, when

Table 5: Resnet architecture. Conv2D(A,B,C) applies a 2d convolution to input with B channels and produces an output with C channels using stride (1, 1), padding (1, 1) and kernel size (A, A).

| Name | Layer | Res. Layer |
|---|---|---|
| Block 1 | Conv2D(3,3,F) LeakyRELU(0.2) Conv2D(3,F,F) AvgPool2D(2) | AvgPool2D(2) Conv2D(1,3,F) no padding |
| | Sum | |
| Block 2 | LeakyRELU(0.2) Conv2D(3,F,F) LeakyRELU(0.2) Conv2D(3,F,F) AvgPool2D(2) | |
| Block 3 | LeakyRELU(0.2) Conv2D(3,F,F) LeakyRELU(0.2) Conv2D(3,F,F) | |
| Block 4 | LeakyRELU(0.2) Conv2D(3,F,F) LeakyRELU(0.2) Conv2D(3,F,F) AvgPool2D(all) | |

both losses in the *CPLearn* objective are minimized these two failure modes are avoided. Additionally, we investigate dimensional collapse following the methodology proposed in previous work (Jing et al., 2022) by computing the singular value distribution of the covariance matrix for the embeddings. In Fig. 13a, we observe that for the undercomplete setting only 10 singular values have large values. This is explained by the fact that embeddings align with the 10 codes in the dictionary, as predicted by Theorem 1. When $c$ increases, more and more singular values increase in their value. This provides evidence that the loss function in conjuction with the proposed projector allows to exploit the whole embedding space and avoid dimensional collapse. Finally, we propose to study intracluster collapse by estimating the entropy of the distribution for the representations. We do so by (i) fitting a Gaussian mixture model with diagonal covariance on the representation from the backbone, (ii) estimating the entropy of the distribution through Monte Carlo using $10k$ samples and (iii) repeating the analysis for different number of mixture components, i.e. $\{10, 20, 50, 100, 200, 500, 1000\}$. Intra-cluster collapse is avoided when achieving higher values of entropy. We illustrate this in Fig. 13b, showcasing improved performance for larger values of dictionary size. We provide additional results for the collapses on SVHN and CIFAR100. Specifically, in Fig. 14 we show the analysis of dimensional collapses, whereas in Fig. 15 we show the one for intracluster collapse.

## L. Experimental Details and Additional Analysis with ResNet-18

**Experimental details for ResNet-18.** We used a ResNet-18 backbone network on CIFAR-10 and train it for 1000 epochs with Adam optimizer, learning rate equal to $1e-3$ and batch-size equal to 64 on 1 A100 GPU. Beta is selected from a smaller subset of values $\{0.1, 0.25, 0.5, 1\}$ (given the more expensive nature of the experiments) to ensure both losses are minimized and chosen being equal to 1.

**Training convergence.** We provide a comparison of the linear probing performance over training on the experiments for ResNet-18 on CIFAR-10. We visualize the top1 accuracy over training epochs both on linear and logarithmic scales. We observe that *CPLearn* achieves better convergence rate compared to the other approaches, thanks to the quasi-orthogonality condition of the code matrix $W$. Results are shown in Fig. 16.

**Time/storage analysis.** In Table 8 we compare all methods in terms of storage (MB) and training time (minutes) using a ResNet-18 trained for 1000 epochs on CIFAR-10. We also provide the linear classification and clustering results for the sake of completeness. The training time does not seem to be impacted much by the larger projector. The size of the checkpoints

Table 6: Hyperparameters (in terms of optimizer and data augmentation) used in SVHN, CIFAR-10 and CIFAR-100 experiments.

| Class | Name param. | SVHN | CIFAR-10 | CIFAR-100 |
|---|---|---|---|---|
| | Color jitter prob. | 0.1 | 0.1 | 0.1 |
| | Gray scale prob. | 0.1 | 0.1 | 0.1 |
| Data augment. | Random crop | Yes | Yes | Yes |
| | Additive Gauss. noise (std) | 0.03 | 0.03 | 0.03 |
| | Random horizontal flip | No | Yes | Yes |
| | Batch size | 64 | 64 | 64 |
| | Epochs | 20 | 200 | 200 |
| Optimizer | Adam $\beta_1$ | 0.9 | 0.9 | 0.9 |
| | Adam $\beta_2$ | 0.999 | 0.999 | 0.999 |
| | Learning rate | $1e-4$ | $1e-4$ | $1e-4$ |

Table 7: Values of $\beta$ hyperparameter. This is chosen from the range $\{0.01, 0.05, 0.1, 0.25, 0.5, 1, 2.5, 5, 10\}$ to ensure that both losses are minimized.

| Dictionary Size | 10 | 128 | 256 | 512 | 1024 | 2048 | 4096 | 8192 | 16384 |
|---|---|---|---|---|---|---|---|---|---|
| SVHN | 0.5 | 0.5 | 0.25 | 0.25 | 0.1 | 0.1 | 0.1 | 0.05 | 0.05 |
| CIFAR-10 | 0.5 | 0.5 | 0.25 | 0.25 | 0.1 | 0.1 | 0.1 | 0.05 | 0.05 |
| CIFAR-100 | 0.5 | 0.5 | 0.25 | 0.25 | 0.1 | 0.1 | 0.1 | 0.05 | 0.05 |

| Method | Clustering | Linear | State dict size [MB] | Full checkpoint size [MB] | Training time [min] |
|---|---|---|---|---|---|
| Barlow | 29.1 | 92.2 | 79 | 157 | 356 |
| SwAV | 18.9 | 89.6 | 53 | 102 | 405 |
| GEDI *no gen* | 44.6 | 80.0 | 47 | 140 | 353 |
| Self-Classifier ($c = f$) | 36.9 | 84.8 | 51 | 144 | 353 |
| Self-Classifier (16384) | 33.9 | 64.9 | 175 | 268 | 355 |
| *CPLearn* ($c = f$) | 47.4 | 91.6 | 51 | 144 | 357 |
| *CPLearn* (16384) | 48.2 | 91.3 | 175 | 268 | 358 |

Table 8: Comparison of methods on clustering and linear evaluation, with model sizes and training times.

do increase significantly. However, the relative difference would be much smaller with larger networks such as ViT-S/B/L or larger ResNet architectures as the backbone itself would be much larger compared to the additional parameters in the projector.

| $\beta$ | Linear |
|---|---|
| 2.0 | 10.0 (cluster collapse) |
| 1.5 | 10.0 (cluster collapse) |
| 1.25 | 10.0 (cluster collapse) |
| 1.0 | 91.6 |
| 0.75 | 90.2 |
| 0.5 | 88.1 |
| 0.25 | 82.7 |
| 0.1 | 75.1 |

Table 9: Effect of varying $\beta$ on linear evaluation performance.

**Sensitivity analysis (on $\beta$).** We conducted additional experiments to evaluate the performance of *CPLearn* across a finer

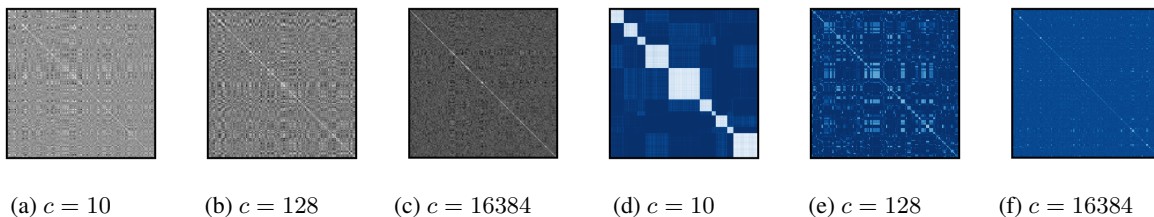

| (a) $c = 10$ | (b) $c = 128$ | (c) $c = 16384$ | (d) $c = 10$ | (e) $c = 128$ | (f) $c = 16384$ |

Figure 8: Realization of embedding covariance (**left**) and adjacency matrices (**right**) for the whole SVHN test dataset. Increasing $c$ reduces the value of the off-diagonal elements of the covariance, thus contributing to increase the decorrelation of features (cf. Corollary 2). Moreover, increasing $c$ has the effect to reduce the block sizes of the adjacency matrix (cf. Corollary 3).

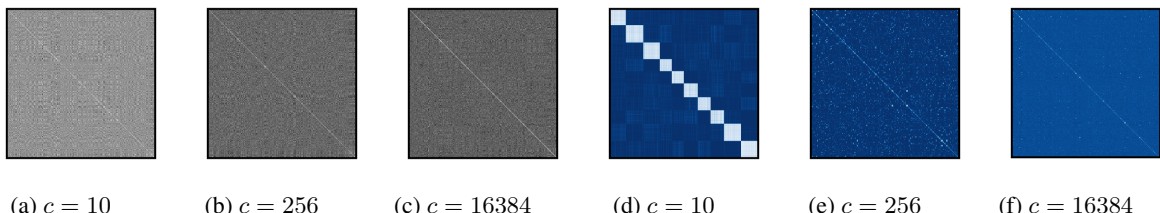

| (a) $c = 10$ | (b) $c = 256$ | (c) $c = 16384$ | (d) $c = 10$ | (e) $c = 256$ | (f) $c = 16384$ |

Figure 9: Realization of embedding covariance (**left**) and adjacency matrices (**right**) for the whole CIFAR-100 test dataset. Increasing $c$ reduces the value of the off-diagonal elements of the covariance, thus contributing to increase the decorrelation of features (cf. Corollary 2). Moreover, increasing $c$ has the effect to reduce the block sizes of the adjacency matrix (cf. Corollary 3).

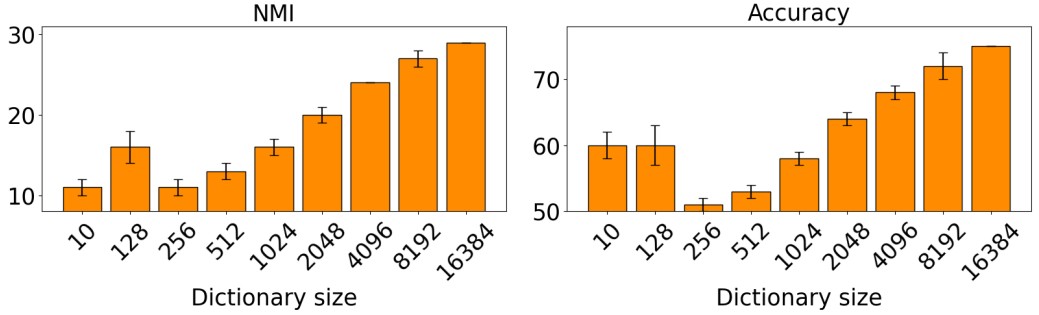

Figure 10: Analysis of downstream generalization for different values of dictionary size on SVHN dataset.

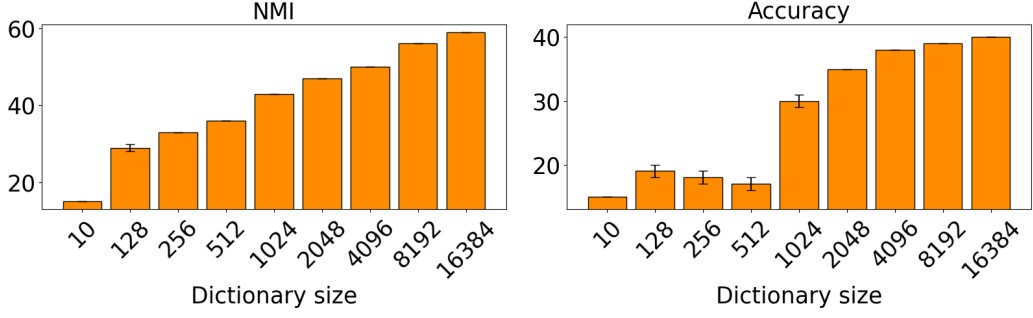

Figure 11: Analysis of downstream generalization for different values of dictionary size on CIFAR-100 dataset.

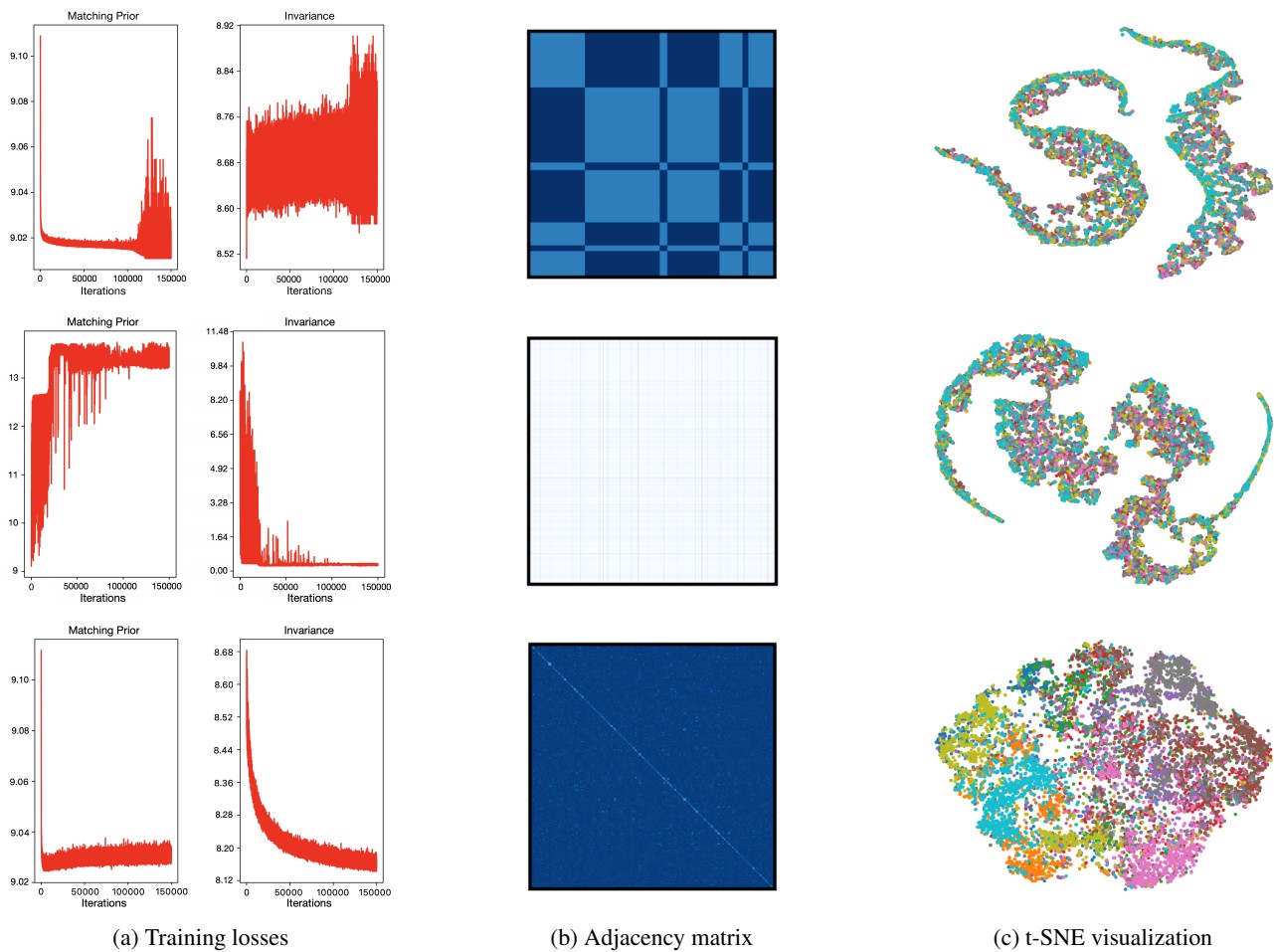

(a) Training losses  (b) Adjacency matrix  (c) t-SNE visualization

Figure 12: Example of bad (**top** with $\beta = 0$ and **middle** with $\beta = 10$) and well-behaved (**bottom**, $\beta = 0.1$) training loss dynamics on CIFAR-10 with dictionary size 8192. When only one term is minimized, the model faces cluster collapse, as demonstrated by the adjacency plots in the top and middle rows (corresponding to 2 and 1 clusters, respectively). However, when both losses are minimized the collapse is avoided. Interestingly, the visualization of the representations reveals the absence of representation collapse in all cases (colors are used to denote different ground truth classes).

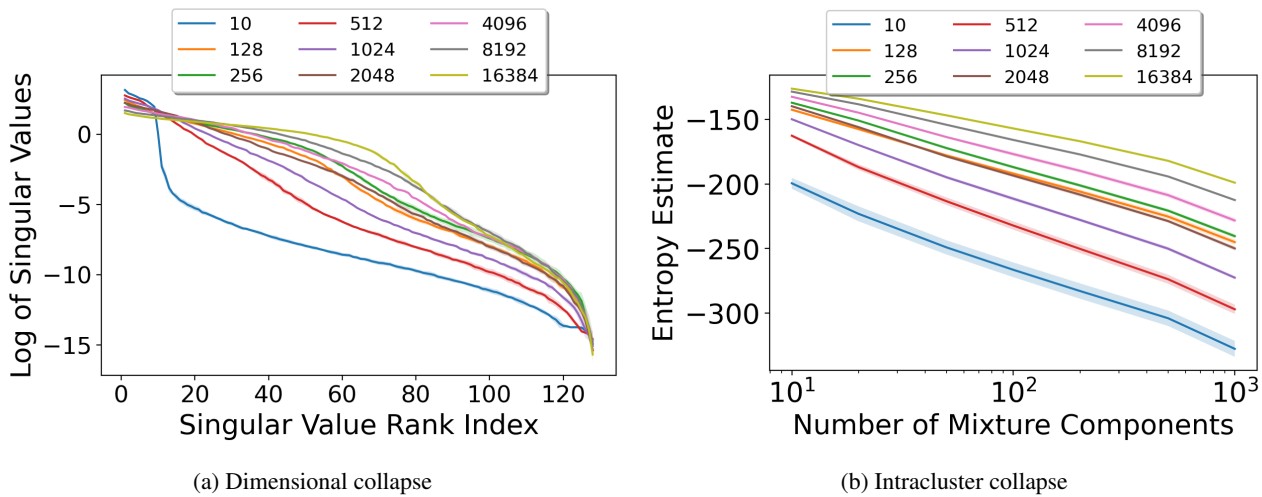

(a) Dimensional collapse

(b) Intracluster collapse

Figure 13: Collapse analysis on CIFAR-10 test data for different dictionary sizes $c$. Results are averaged over 5 training runs obtained from random initialization seeds. **Left:** The singular values of the embedding covariance are in sorted order and logarithmic scale. The curve rises with very large values of $c$, avoiding zero singular values. **Right:** The number of mixture components are in logarithmic scale. The curve rises with very large values of $c$ for all number of mixture components.

range of $\beta$ values, using ResNet-18 on CIFAR-10. The results are presented in Table 9. We observed that performance improves as $\beta$ increases. However, excessively large values of $\beta$ lead to cluster collapse. This occurs because minimizing only the invariance loss ensures invariance and low-entropy predictions, but does not guarantee satisfaction of the matched prior condition described in Lemma 1. In other words, the network tends to use only a subset of codes, resulting in cluster collapse.

## M. Experimental Details on ImageNet-100

**Training.** We used a ViT-small backbone network and train it for 100 epochs with learning rate equal to $5e - 4$ and batch-size per GPU equal to 64 on a node with 8 NVIDIA A100 GPUs. Beta is selected from a smaller subset of values $\{0.1, 0.25, 0.5, 1\}$ (given the more expensive nature of the experiments) to ensure both losses are minimized and chosen being equal to 0.25.

**Evaluation.** For linear probe evaluation, we use the DINO codebase and train the classifier with Adam optimizer (Caron et al., 2021).

## N. Practical Implementation of the Loss

We observed training instability when using the larger backbone on ImageNet-100. The issue is due to some dictionary codes being unused during the initial training phase (cluster collapse), making the KL matching prior term infinity. Indeed, we have that

$$\mathcal{L}_{CPLearn}(\mathcal{D}) = \beta CE(\boldsymbol{p}, \boldsymbol{p}') + CE(\boldsymbol{q}, \boldsymbol{p})$$
$$\propto \beta CE(\boldsymbol{p}, \boldsymbol{p}') + KL(\boldsymbol{q}, \boldsymbol{p})$$

In practice, the reverse KL term is sufficient to avoid the issue:

$$\mathcal{L}_{CPLearn}(\mathcal{D}) = \beta CE(\boldsymbol{p}, \boldsymbol{p}') + KL(\boldsymbol{p}, \boldsymbol{q})$$

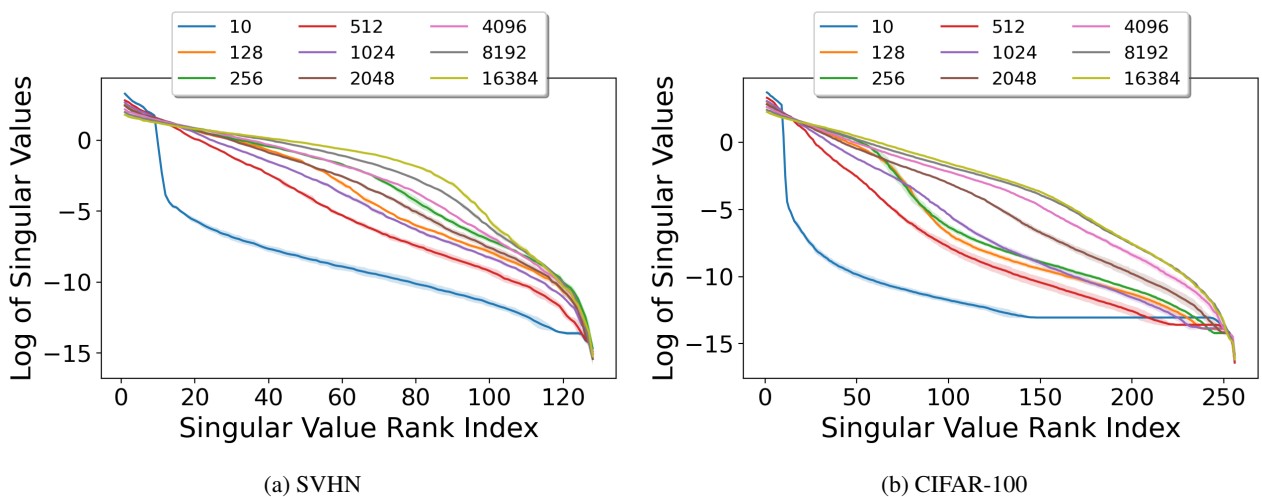

(a) SVHN

(b) CIFAR-100

Figure 14: Dimensional collapse analysis on test data for different size of dictionary. Results are averaged over 5 training runs obtained from random initialization seeds.

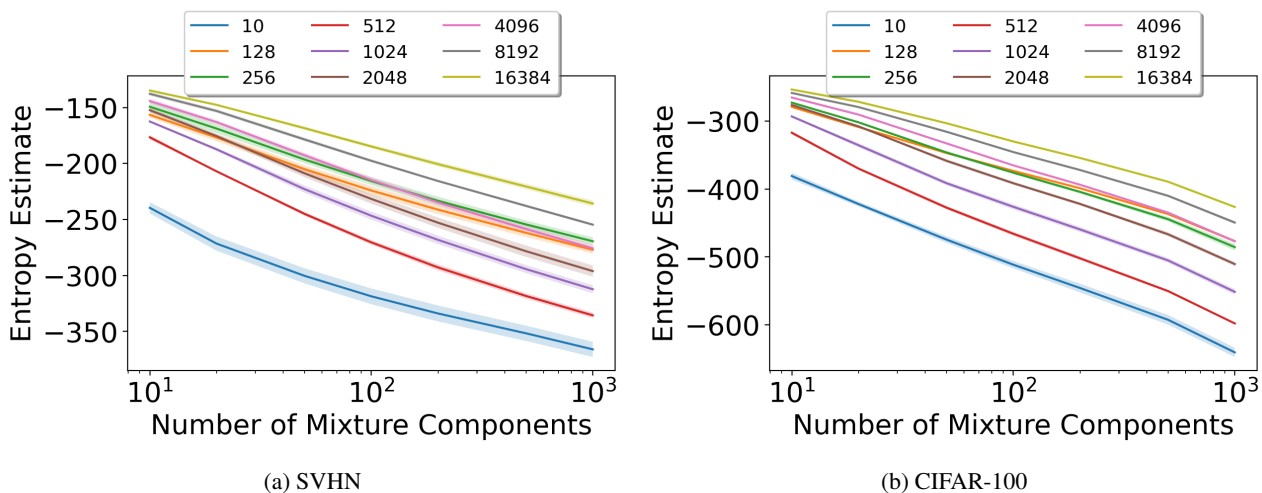

(a) SVHN

(b) CIFAR-100

Figure 15: Intracluster collapse analysis on test data for different size of dictionary. Results are averaged over 5 training runs obtained from random initialization seeds.

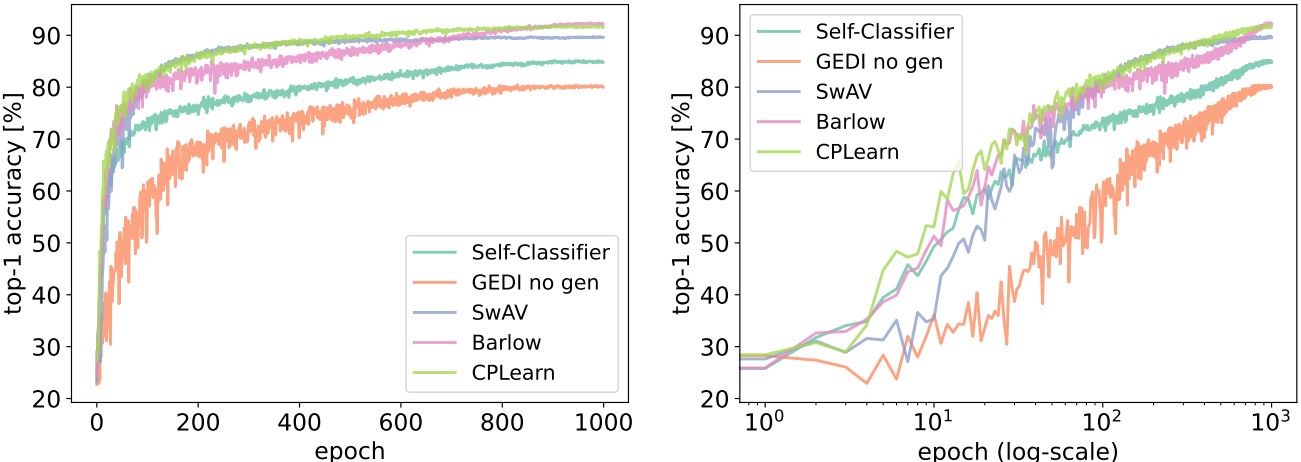

Figure 16: Analysis of training convergence.

