# OpenReview forum: "Collapse-Proof Non-Contrastive Self-Supervised Learning"
_ICML.cc/2025/Conference — ICML 2025 poster_

### Official Review · Reviewer_MTDo · 2025-03-14

**Overall Recommendation:** 4

**Summary:**

This work studies non-contrastive self-supervised representation learning. The authors identify known collapse modes in non-contrastive SSL and propose a collapse-proof approach (CPLearn). They show that CPLearn jointly decorrelates and clusters embeddings, avoiding common collapse modes without the need for heuristics as in previous literature.

**Claims And Evidence:**

To support the claims, the paper provides mathematical proofs to specific design choices such as the loss function and the projector, and provides experiment results on four classic SSL image datasets. The proposed method (CPLearn) achieves competitive performance when the dictionary size gets large.

**Essential References Not Discussed:**

None.

**Experimental Designs Or Analyses:**

Experimental designs and analyses are sound. Besides the experiments, visualization on important aspects of the learning process are also provided in Figure 3 and 4, which are informative with detailed explanations.

**Methods And Evaluation Criteria:**

The method essentially projects and soft assigns learned representations to a code book and computes the loss w.r.t the codes. The design intuitively makes sense and is supported by the provided theoretical justifications.

The evaluation criteria adopt standard SSL evaluation. Experiments are conducted on four classic SSL image datasets and compared with representative baselines in the field. The practical choice of a critical hyperparameter (c) is also discussed and experimented. Additionally, the performance range for multiple runs is provided in the tables, making the evaluation solid.

**Other Comments Or Suggestions:**

None.

**Other Strengths And Weaknesses:**

Strengths:
1. Theories rather than heuristics are presented to support the design choices and are backed by experiment results and visualization.
2. The proposed method seems straightforward to implement. The provided pseudocode is clear.

Weaknesses:
See questions.

**Questions For Authors:**

Vicreg is also a mainstream non-contrastive SSL method, why was it not compared with in the experiments?

Results on ImageNet are not provided. Are there any difficulties in applying the method to a larger scale of SSL training, for example, on ImageNet?

How fast is the training convergence compared with the baselines?

**Relation To Broader Scientific Literature:**

The paper provides sufficient literature discussion.

**Theoretical Claims:**

I did not rigorously check the correctness of the proofs.

---

> ### Author Rebuttal · Authors · 2025-04-01
>
> Thank you for the appreciation of our work and the time dedicated to review our paper. Please find below the answers to your questions:
>
> **Comparison with VicReg** We agree that VicReg is another non-contrastive self-supervised method. Given the limited amount of time available for the rebuttal, we have decided to provide additional results including experiments with a ResNet-18 on CIFAR10 and add additional baselines, such as SwAV and GEDI no gen, on ImageNet-100 to answer the requests from other reviewers (please refer to the answer to Reviewer KMvq). While we didn’t perform an explicit comparison with VicReg on ResNet-8 and ImageNet-100, it is still possible to observe that VicReg achieves similar performance to Barlow Twins when using a Resnet-18 on CIFAR-10 by looking at the results reported in the [solo-learn](https://github.com/vturrisi/solo-learn) repository, which is used in our experiments (e.g. Barlow Twins 92.1% vs. VicReg 92.1% acc@1 on CIFAR-10). This is reasonable as VicReg belongs to the family of feature decorrelation approaches like Barlow Twins.
>
> **Training Full ImageNet** The main bottleneck arises from the fact that we don’t have access to large computational resources to scale up to full ImageNet.
>
> **Training convergence** We provide a comparison of the linear probing performance over training on the experiments for ResNet-18 on CIFAR10. We visualize the top1 accuracy over training epochs both on linear and logarithmic scales. We observe that CPLearn achieves better convergence rate compared to the other approaches, thanks to the quasi-orthogonality condition of the code matrix W. The plots are available at the following anonymous [link](https://aquamarine-lula-78.tiiny.site) and will be added in the Supplementary Material together with the additional experiments done in this rebuttal (that is experiments with a ResNet-18 on CIFAR10 and additional baselines, such as SwAV and GEDI no gen, on ImageNet-100).
>
> Please let us know if you have any additional questions.

---

> > ### Comment · Reviewer_MTDo · 2025-04-09
> >
> > Thank you for the rebuttal.
> >
> > I have read the other rebuttals and the additional results, and I encourage adding these results to the manuscript during revision to make the paper more complete. I appreciate the plots on convergence speed.  Overall, the paper puts emphasis on its theoretical motivation, so I am okay with the smaller-scale experiments and some of the results being comparable to baselines. I am willing to keep my original rating, and I am looking forward to seeing more evaluations on how pretrained CPLearn can perform on downstream vision tasks in the future.

---

> > > ### Author Response · Authors · 2025-04-09
> > >
> > > We appreciate reviewer’s engagement and the additional feedback. Indeed, we will include all analyses and discussions in the final paper, as they all have contributed to enhance the completeness of the work.
> > >
> > > Given the allowed extra page, we will do the following modifications to the main text:
> > > 1. Add the discussion about the choice of Rademacher distribution for the hyperdimensional computing vectors
> > > 2. Include the additional baselines (SwAV and GEDI no gen) for the experiments on ImageNet-100
> > > 3. Add the extended analysis about dimensional and intra-cluster collapses for all methods -> see [Figure](https://anonymous.4open.science/r/ICML12199-1261/Collapses.pdf)
> > >
> > > In the Supplementary, we will include all results from the rebuttal:
> > > 1. The analysis of the training convergence of all methods -> [Figure](https://anonymous.4open.science/r/ICML12199-1261/Convergence.pdf)
> > > 2. The experiments on Resnet-18
> > > 3. The time/storage analysis of all methods
> > > 4. The analysis of the $\beta$ hyperparameter for CPLearn
> > >
> > > Thank you once again for the interaction.

---

### Official Review · Reviewer_vWmQ · 2025-03-14

**Overall Recommendation:** 3

**Summary:**

This paper theoretically demonstrate the conditions of avoiding four kinds of collapse in non-contrastive self-supervised learning based on the CPLearn projector design. Specifically, the authors prove that minimizing invariance to data augmentations while matching priors suffices to avoid representation and cluster collapses, whereas orthogonal frozen weights based on hyperdimensional computing, and large prediction outputs in the projector are key to avoiding dimensional and intracluster collapses. The theoretical claims are validated through experiments on benchmark datasets.

**Claims And Evidence:**

Yes, the claims are well supported.

**Essential References Not Discussed:**

NA.

**Experimental Designs Or Analyses:**

Please refer to *Methods And Evaluation Criteria*.

**Methods And Evaluation Criteria:**

- The NMI and accuracy of the proposed method are tested on several benchmark datasets, whereas in Tables 1 and 2 the selection of compared methods are different. I think the authors should explain why.
- The experiments are conducted on small benchmark datasets. Additional results on e.g. full ImageNet would be more convincing.
- The accuracy improvements in Table seem marginal.

**Other Comments Or Suggestions:**

1. The study of optimal $\beta$ parameter is lacking.
2. Typo in Lemma 1. $q_{ij}$ should be $q_j$?

**Other Strengths And Weaknesses:**

**Strengths**
- The paper is overall well written with the claims well supported.
- This paper established the sufficient condition for avoiding collapse through sound theoretical analysis.

**Weaknesses**
- The intuitive explanations of the theoretical settings and results could be enhanced, e.g. why is the W matrix assumed to follow the Rademacher distribution?
- The experimental are relatively small scale, but this should not be over criticized since this paper mainly focus on the theoretical perspectives.

**Questions For Authors:**

1. Why is W assumed to follow the Rademacher distribution?
2. Why do Table 1 and Table 2 use different compared methods?
3. How much extra computational burden does the large projector bring? e.g. extra computing time/storage?
4. How do the dimensional and intracluster collapses of CPLearn compared with related methods? (Figures 12-14)

**Relation To Broader Scientific Literature:**

NA.

**Theoretical Claims:**

- I skimmed through the proofs but did not check the details. They seem to make sense.
- Line 155 (left). The matrix W is drawn from the Rademacher distribution. Is there any explanations on this assumption? I suppose in practice the W matrix is learned through optimization instead of drawn from a certain distribution.

---

> ### Author Rebuttal · Authors · 2025-04-01
>
> Thank you for appreciating the theoretical nature of our paper and providing constructive suggestions.
>
> **Baselines for Table 1 and Table 2** Thank you for the suggestion. We did an extra effort in the limited time available to provide additional baselines and make Table 2 more complete. Please find below the table of results with the addition of GEDI no gen (the version without the generative model) and SwAV. The best performance is highlighted in bold for each method. Overall, we observe that CPLearn achieves linear probing performance comparable to the best methods, such as SwAV and Barlow. Moreover, CPLearn significantly outperforms all other approaches in terms of clustering performance. This stems from the unique capability of the model to perform both clustering and feature decorrelation.
>
> | | **Clust** |  |  |  |  |  | **Linear** |  |  |  |  |  |
> | - | - | - | - | - | - | - | - | - | - | - | - | - |
> | **Projector** | **Small** | | | **Large** | | | **Small** | | | **Large** | | |
> | **Method**                | **128**   | **1024**  | **2048**  | **32768** | **65536** | **131072** | **128**   | **1024**  | **2048**  | **32768** | **65536** | **131072** |
> | Barlow           | 30.0% | 53.6% | **59.4%** | oom | oom | oom | 66.9% | **77.2%** | **77.2%** | oom | oom | oom |
> | SwAV             | 28.0% | 47.9% | 51.2% | 60.7% | 60.5% | **62.8%** | 76.6% | 77.7% | **78.0%** | 76.6% | 77.7% | 77.3% |
> | DINO             | 46.0% | 53.4% | 55.2% | 63.1% | **64.7%** | 64.3% | 71.8% | 73.6% | 73.9% | 75.1% | **76.2%** | 75.8% |
> | GEDI no gen    | 24.5% | 36.3% | **38.8%** | 32.7% | 32.9% | 33.3% | 71.8% | **73.2%** | 72.8% | 72.9% | 72.7% | 72.8% |
> | CPLearn           | 34.1% | 54.3% | 57.9% | 69.3% | 69.6% | **70.9%** | 70.8% | 72.8% | 70.1% | 74.5% | 74.7% | **78.0%** |
>
> **Clarity: choice of Rademacher distribution** Thank you for the curious question. There are several ways to define random code vectors in hyperdimensional computing (HC). We chose to use the multiply-add-permute encoding, which leverages Rademacher distribution to sample them. We refer you to a recent survey on HC for more details [1]. This is the simplest form of encoding equipped with simple element-wise addition and multiplication operations to perform algebraic compositions. We are going to add this in the paper and reference the survey. It is also important to mention that the exploitation of the compositional properties of HC is beyond the scope of the paper, but we are actively working on this direction. \
> Also, it is important to clarify that the code matrix W is fixed throughout training. This is necessary to achieve the collapse-proof property as demonstrated by the theory in the paper (cf. Theorem 1 and related Corollaries).
>
> **Refer**  \
> [1] Kleyko et al. A Survey on Hyperdimensional Computing aka Vector Symbolic Architectures, Part I: Models and Data Transformations. ACM Comp. Surv. 2022
>
> **Time/storage analysis** We provide a table for the experiments with ResNet-18 trained for 1000 epochs on CIFAR10 comparing all methods in terms of storage (MB) and training time (minutes). We also provide the linear classification and clustering results for the sake of completeness. The training time does not seem to be impacted much by the larger projector. The size of the checkpoints do increase significantly. However, the relative difference would be much smaller with larger networks such as ViT-S/B/L or larger ResNet architectures as the backbone itself would be much larger compared to the additional parameters in the projector.
>
> | Method                    | Clust | Linear | State dict size [MB] | Full checkpoint size [MB] | Train time [min] |
> | - | - | - | - | - | - |
> | Barlow                    | 29.1       | 92.2   | 79                   | 157                       | 356                 |
> | SwAV                      | 18.9       | 89.6   | 53                   | 102                       | 405                 |
> | GEDI no gen               | 44.6       | 80     | 47                   | 140                       | 353                 |
> | Self-Classifier (c=10)    | 54.2       | 78     | 47                   | 140                       | 347                 |
> | Self-Classifier (c=512)   | 36.9       | 84.8   | 51                   | 144                       | 353                 |
> | Self-Classifier (c=16384) | 33.9       | 64.9   | 175                  | 268                       | 355                 |
> | CPLearn (c=10)            | 62.1       | 85.4   | 47                   | 140                       | 350                 |
> | CPLearn (c=512)           | 47.4       | 91.6   | 51                   | 144                       | 357                 |
> | CPLearn (c=16384)         | 48.2       | 91.3   | 175                  | 268                       | 358                 |
>
> **Typo Lemma 1** Indeed that is a typo. Thank you for spotting it !
>
> We hope we've effectively addressed your concerns and encouraged you to reconsider your rating.

---

> > ### Comment · Reviewer_vWmQ · 2025-04-05
> >
> > I appreciate the authors' reply, but some of my concerns seem to be overlooked, e.g., the parameter study of $\beta$ and the collapse analysis compared with related works.

---

> > > ### Author Response · Authors · 2025-04-06
> > >
> > > Thank you for the additional feedback. Given the space limitations of the rebuttal and the time available, we have decided to defer answering some of the questions to a second stage. Please find detailed responses to the remaining questions below.
> > >
> > > **Analysis of $\beta$** We conducted additional experiments to evaluate the performance of CPLearn across a finer range of $\beta$ values, using ResNet-18 on CIFAR-10. The results are presented in the table below. We observed that performance improves as $\beta$ increases. However, excessively large values of $\beta$ lead to cluster collapse. This occurs because minimizing only the invariance loss ensures invariance and low-entropy predictions, but does not guarantee satisfaction of the matched prior condition described in Lemma 1. In other words, the network tends to use only a subset of codes, resulting in cluster collapse. We will include this additional analysis in the Appendix of the final paper.
> > >
> > > | $\beta$ | linear                |
> > > | - | - |
> > > | 2    | 10 (cluster collapse) |
> > > | 1.5  | 10 (cluster collapse) |
> > > | 1.25 | 10 (cluster collapse) |
> > > | 1    | 91.6                  |
> > > | 0.75 | 90.2                  |
> > > | 0.5  | 88.1                  |
> > > | 0.25 | 82.7                  |
> > > | 0.1  | 75.1  |
> > >
> > > **Dimensional and intra-cluster collapse analysis for different methods** Thank you for the insightful and constructive suggestion. We have extended the analysis in Figs. 12–14 to include a comparison of different methods on CIFAR-10 and CIFAR-100. Since different SSL methods have different representation statistics, we standardize the representations (zero mean and unit standard deviation) to allow for a comparison. All results are available at the following anonymous [link](https://anonymous.4open.science/r/ICML12199-1261/Collapses.pdf). Reported metrics are different to those in Figure 12-14 (where no standardization has been used). Overall, we observe that CPLearn is the only method that remains robust to both dimensional and intracluster collapse across both datasets. These results will be included in the Appendix of the final paper. Thank you again for suggesting this valuable analysis !
> > >
> > > Please let us know if you have any additional questions.

---

### Official Review · Reviewer_nCR1 · 2025-03-14

**Overall Recommendation:** 2

**Summary:**

This paper introduces CPLearn, a novel self-supervised non contrastive approach that avoids heuristics such as stop gradient or momentum encoders for feature collapse. CPLearn does this by utilizing a projector module and a special loss function which minimizes the invariance between augmented views while enforcing a prior distribution (uniform) on the features. The authors evaluated their approach on multiple different datasets, such as SVHN, CIFAR-10, CIFAR-100, and ImageNet100.

**Claims And Evidence:**

The experiments are conducted in a non-standard manner, thus hardly justifying if the experimental results well support the claims.

**Essential References Not Discussed:**

There are not related works that are essential to understanding the key contributions which are missing.

**Experimental Designs Or Analyses:**

The experiments are sound but some of the metrics and training configurations do not necessarily match those of previous works. Specifically, previous works typically do not use ResNet 8 or ViT models as the backbone. NMI is also not a metric typically used which makes comparison with other SSL models difficult.

**Methods And Evaluation Criteria:**

The methods and evaluation criteria make sense for this application.

**Other Comments Or Suggestions:**

The paper was well written overall, but there seems to be a grammatical issue at the end of the “Generalization on downstream tasks” subsection in the sentence “…Self-Classifier and GEDI achieve overall perform better on clustering tasks.”

**Other Strengths And Weaknesses:**

This paper is well-written, clearly organized, and effectively structured, with figures and algorithmic descriptions that enhance the understanding of CPLearn. The theoretical foundations are solid, and the approach is rigorously validated across multiple datasets. However, since CPLearn leverages hyperdimensional computing, a discussion on computational resource usage and potential scalability concerns would strengthen the paper since the instability observed when using larger backbones raises concerns about the practicality of the method in large-scale settings. Providing a justification for why CPLearn remains valuable despite this limitation, or exploring possible solutions, would improve the overall discussion.

**Questions For Authors:**

No questions

**Relation To Broader Scientific Literature:**

As self-supervised models have been increasingly researched, many have utilized heuristics to address problems of collapse. This paper introduces a new approach to circumvent this issue and provides mathematical justification to support it can address collapse.

**Theoretical Claims:**

I did not check the correctness of any proofs.

---

> ### Author Rebuttal · Authors · 2025-04-01
>
> Thank you for appreciating the theoretical nature of our paper and for the time dedicated to review it. Please find below the answers to the major concerns.
>
> **Experiments on ResNet-18** Thank you for the suggestion. Please find below the table with the results on CIFAR10 using ResNet-18. We used the [solo-learn](https://github.com/vturrisi/solo-learn) codebase to run the analysis with the same suggested hyperparameter configurations. Note that we reproduced all baselines ourselves and that their performance are slightly higher than what is reported in [solo-learn](https://github.com/vturrisi/solo-learn). Overall, the table highlights similar observations to the ones from experiments on ResNet-8, with CPLearn achieving comparable performance to Barlow Twins and significantly outperforming all other approaches in terms of clustering. Interestingly, both Self-Classifier and CPLearn achieve the highest clustering performance for $c=10$, corresponding to the ground truth number of classes.
>
> | Method                    | Clustering | Linear |
> | - | - | - |
> | Barlow                    | 29.1       | **92.2**   |
> | SwAV                      | 18.9       | 89.6   |
> | GEDI no gen               | 44.6       | 80.0     |
> | Self-Classifier (c=10)    | 54.2       | 78.0     |
> | Self-Classifier (c=512)   | 36.9       | 84.8   |
> | Self-Classifier (c=16384) | 33.9       | 64.9   |
> | CPLearn (c=10)            | **62.1**       | 85.4   |
> | CPLearn (c=512)           | 47.4       | **91.6**   |
> | CPLearn (c=16384)         | 48.2       | 91.3   |
>
> **Experiments with ViT on ImageNet-100** We respectfully disagree with the reviewer. ViT is a standard backbone for benchmarking self-supervised learning algorithms, see for instance [DINO](https://github.com/facebookresearch/dino).
>
> **Typos** Thank you for spotting the typos. Indeed, the sentence “whereas SwAV, Self-Classifier and GEDI achieve overall perform better on clustering tasks” should be rephrased to “​​whereas SwAV, Self-Classifier and GEDI perform better on clustering tasks than Barlow Twins.”
>
> We hope we've effectively addressed your concerns and encouraged a reconsideration of your rating.

---

### Official Review · Reviewer_KMvq · 2025-03-14

**Overall Recommendation:** 3

**Summary:**

The paper introduces CPLearn, a non-contrastive self-supervised learning method designed to avoid common failure modes—namely, representation, dimensional, cluster, and intracluster collapses. The authors propose a simple projector design and loss function, leveraging ideas from hyperdimensional computing, that naturally encourage both decorrelation and clustering of embeddings without relying on heuristics like stop gradient or momentum encoders. The method is theoretically analyzed, and its properties are established through several results. Experimental results are conducted on datasets such as SVHN, CIFAR-10, CIFAR-100, and ImageNet-100 to test for robust generalization in clustering and linear classification tasks.

**Claims And Evidence:**

The claims are supported by theoretical proofs (Lemma 1, Theorem 1) and empirical validation. The proofs leverage convex optimization and quasi-orthogonality assumptions, and experiments demonstrate improved generalization and collapse avoidance. However, the assumption that $W^TW\approx fI$ for large $c$ (practical quasi-orthogonality) is not rigorously validated beyond empirical histograms (Fig. 3). Further, the experiments use a ResNet-8 backbone on CIFAR datasets, resulting in lower overall performance compared to more standard architectures (e.g., ResNet-18/ResNet-50).

**Essential References Not Discussed:**

N/A

**Experimental Designs Or Analyses:**

The experimental design is generally sound, with evaluations conducted on several datasets and extensive ablation studies. The analysis of dictionary size effects (using different settings for c) and the comparisons against multiple baselines are strengths. However, the use of a ResNet-8 backbone on CIFAR datasets leads to overall low performance compared to more common architectures, making it unclear whether the improvements would hold in more standard settings. Including experiments with ResNet-18 and adding baselines for larger-scale settings like ImageNet-100 would provide a more convincing picture.

**Methods And Evaluation Criteria:**

The methods are novel and interesting: the projector design and loss terms directly address collapse modes, and the use of large dictionaries aligns with theoretical insights. Evaluation on standard SSL benchmarks (CIFAR, ImageNet-100) is appropriate, and metrics (NMI, linear probe accuracy) are widely accepted. Nonetheless, including experiments with more standard backbones like ResNet-18 and additional baselines on ImageNet-100 would improve the empirical analysis

**Other Comments Or Suggestions:**

N/A

**Other Strengths And Weaknesses:**

Strengths:

- The paper provides an in-depth analysis of clustering-related collapse, extending prior work that mainly focused on dimensional collapse.

- It offers a principled, unified design that tackles multiple collapse modes simultaneously.

- The related works section is comprehensive and well-informed, covering a broad range of literature.

Weaknesses:

- The experiments on CIFAR-10/100 use a ResNet-8 backbone, which leads to performance levels that are significantly lower (about 25%) than what is typically observed with architectures like ResNet-18. Its unclear how these gains translate to standard and more widely used architectures.

- The paper omits baseline comparisons on ImageNet-100 with methods such as GEDI and SwAV, leaving it unclear how CPLearn performs in larger-scale settings compared to the closest baselines.

- Based on the ImageNet-100 results, CPLearn does not consistently outperform DINO or Barlow Twins. The projection sizes at which CPLearn shows an advantage are not reported for some of these baselines, as many values are missing due to out-of-memory errors.

- The paper is very hard to follow. For instance, the introduction doesn't even define representation collapse and isn't well-written and motivated. It gives a very shallow background.

- In the methods section, the authors don't give any insights into their design choices corresponding to $H$ and $P$ matrices. Why is $W$ being sampled from the Rademacher distribution? Similarly, there are no insights around the final loss term. Some of the ways in which the first term promotes invariance can be easily mentioned since it minimizes the effective cross-entropy loss.

**Questions For Authors:**

N/A

**Relation To Broader Scientific Literature:**

The work bridges non-contrastive SSL (e.g., Barlow Twins) and cluster-based SSL (e.g., SwAV), addressing gaps in collapse prevention. It extends Sansone (2023) by removing the generative term and formalizing guarantees. The hyperdimensional computing connection is novel but underdeveloped. The related works section is extensive and well-contextualizes the contributions within a broad range of literature.

**Theoretical Claims:**

Lemma 1’s derivation via Lagrangian multipliers and KKT conditions is sound. Theorem 1 assumes an orthogonal $W$, which is relaxed via hyperdimensional computing principles.

---

> ### Author Rebuttal · Authors · 2025-04-01
>
> Thank you for appreciating our work and the constructive review.
>
> **Validation of quasi-orthogonality** $W^TW=fI$ holds in probabilistic terms, formally governed by Eq. 5 in the paper, namely in expectation we have $E_W[cos(w_i,w_j)]=\delta(i-j)$, with $\delta$ being a Kronecker delta function (equal to 1 iff $i=j$ and zero otherwise), and its variance is determined by $Var_W[cos(w_i,w_j)]=1/f$. For large $f$, the variance quickly approaches zero and the equality holds almost surely.
>
> **Baselines on ImageNet-100** Please find below the table of results with the addition of GEDI no gen and SwAV. The best performance are highlighted in bold for each method. Overall, we observe that CPLearn achieves linear probing performance comparable to the best methods, such as SwAV and Barlow. Moreover, CPLearn significantly outperforms all other approaches in terms of clustering performance. This stems from the unique capability of the model to perform both clustering and feature decorrelation.
>
> | | **Clust** |  |  |  |  |  | **Linear** |  |  |  |  |  |
> | - | - | - | - | - | - | - | - | - | - | - | - | - |
> | **Projector** | **Small** | | | **Large** | | | **Small** | | | **Large** | | |
> | **Method**                | **128**   | **1024**  | **2048**  | **32768** | **65536** | **131072** | **128**   | **1024**  | **2048**  | **32768** | **65536** | **131072** |
> | Barlow           | 30.0% | 53.6% | **59.4%** | oom | oom | oom | 66.9% | **77.2%** | **77.2%** | oom | oom | oom |
> | SwAV             | 28.0% | 47.9% | 51.2% | 60.7% | 60.5% | **62.8%** | 76.6% | 77.7% | **78.0%** | 76.6% | 77.7% | 77.3% |
> | DINO             | 46.0% | 53.4% | 55.2% | 63.1% | **64.7%** | 64.3% | 71.8% | 73.6% | 73.9% | 75.1% | **76.2%** | 75.8% |
> | GEDI no gen    | 24.5% | 36.3% | **38.8%** | 32.7% | 32.9% | 33.3% | 71.8% | **73.2%** | 72.8% | 72.9% | 72.7% | 72.8% |
> | CPLearn           | 34.1% | 54.3% | 57.9% | 69.3% | 69.6% | **70.9%** | 70.8% | 72.8% | 70.1% | 74.5% | 74.7% | **78.0%** |
>
> **Experiments on Resnet-18** Please find below the table with the results on CIFAR10 using ResNet-18. We used the [solo-learn](https://github.com/vturrisi/solo-learn) codebase to run the analysis with the same suggested hyperparameter configurations. Note that we reproduced all baselines ourselves and that their performance are slightly higher than what reported in [solo-learn](https://github.com/vturrisi/solo-learn). Also please note that NMI is a standard and well-established metric to measure clustering performance. Overall, the table highlights similar observations to the ones from experiments on ResNet-8, with CPLearn achieving comparable performance to Barlow Twins and significantly outperforming all other approaches in terms of clustering. Interestingly, both Self-Classifier and CPLearn achieve the highest clustering performance for $c=10$, corresponding to the ground truth number of classes.
>
> | Method                   | Clustering | Linear |
> | - | - | - |
> | Barlow                    | 29.1       | **92.2**   |
> | SwAV                      | 18.9       | 89.6   |
> | GEDI no gen               | 44.6       | 80.0     |
> | Self-Classifier (c=10)    | 54.2       | 78.0     |
> | Self-Classifier (c=512)   | 36.9       | 84.8   |
> | Self-Classifier (c=16384) | 33.9       | 64.9   |
> | CPLearn (c=10)            | **62.1**       | 85.4   |
> | CPLearn (c=512)           | 47.4       | **91.6**   |
> | CPLearn (c=16384)         | 48.2       | 91.3   |
>
> **Clarity about collapses** We are going to move Figure 1 (showcasing all collapses) in the 1st page and then reference it at line 15 to introduce it early on.
>
> **Design choices**. The activation function in $H$ is chosen in a way to ensure the validity of Theorem 1. We have an extra linear layer with batch norm in $H$ to ensure well-behaved statistics throughout training (as shown in Appendix A). $P$ is a standard linear layer with softmax activation. \
> There are several ways to define random code vectors in hyperdimensional computing (HC). We chose to use the multiply-add-permute encoding, which leverages a Rademacher distribution. We refer you to a recent survey on HC for more details [1]. This is the simplest form of encoding equipped with simple element-wise addition and multiplication operations to perform algebraic compositions. We are going to add this in the paper and reference the survey. It is also important to mention that the exploitation of the compositional properties of HC is beyond the scope of the paper, but we are actively working on this direction. \
> We are going to explicitly mention that Eq. 2 consists of a sum of two cross-entropy losses (this can be also seen from Algorithm 1 in the main paper).
>
> We hope we've effectively addressed your concerns and encouraged a reconsideration of your rating.
>
> **Reference** \
> [1] A Survey on Hyperdimensional Computing aka Vector Symbolic Architectures, Part I: Models and Data Transformations. ACM Comp. Surv. 2022

---

### Decision · Program_Chairs · 2025-05-01

**Decision:**

Accept (poster)

**Comment:**

This submission received ratings of 4,3,3,2. The weak reject reviewer likes the overall paper but complains about larger scale experiments. Since the other reviewers are generally positive about the paper, it will be accepted.